# Investigating ice formation pathways using a novel two-moment multi-class cloud microphysics scheme

Tim Lüttmer[1], Peter Spichtinger[1], and Axel Seifert[2]

[1]Institute for Atmospheric Physics, Johannes Gutenberg University Mainz, Mainz, Germany
[2]Deutscher Wetterdienst, Offenbach, Germany

**Correspondence:** Tim Lüttmer (tluettm@uni-mainz.de)

**Abstract.** For pure ice clouds in the cold temperature regime ($T < 235\,\mathrm{K}$), two major formation pathways are possible. Liquid origin ice clouds stem from droplets that freeze close to water saturation. In-situ formed ice clouds form directly from the vapor phase below water saturation. For a better investigation of these pathways, we developed a novel microphysics scheme. The new two-moment scheme distinguishes between five ice classes ('ice modes') each with their unique formation mechanism: homogeneous freezing of solution droplets, deposition nucleation, homogeneous freezing of cloud droplets and raindrops, immersion freezing and secondary ice from rime splintering. The ice modes interact with each other, e.g. in competition for growth by deposition of water vapor and aggregation, but also with the other cloud particle classes, i.e., cloud droplets, rain, snow, graupel, hail.

This scheme was employed to investigate the liquid origin vs in-situ formation in the fully glaciated parts of an idealised convective cloud. The majority of the cloud ice in the deep convection cloud consisted of frozen droplets (liquid origin). This was caused by the high number concentration of cloud droplets available for freezing. In-situ formed ice was only relevant for the overshoot where ice from both formation pathways mixed.

The new scheme is also useful for investigation of the ice formation in the mixed-phase parts of the convective cloud. We find a vertical layering of ice modes in the cloud. The lower most layer consists of secondary ice from rime splintering and occurred near the updraft core at temperatures around the Hallet-Mossop zone. At altitudes between 6 and 9 km, ice mostly stems from immersion freezing. We find a correlation between the abundance of ice from immersion freezing and snow. The majority of ice crystals above 9 km stems from homogeneously frozen cloud droplets since ice nucleating particles (INP) required for immersion freezing where quickly depleted.

## 1 Introduction

Clouds cover a large part of the planet and constitute an important component of the Earth-Atmosphere system. We can discriminate between different thermodynamic regimes for clouds. For temperatures above melting point ($T > 273\,\mathrm{K}$), clouds consist of liquid water droplets. In the temperature regime $235\,\mathrm{K} \leq T \leq 273\,\mathrm{K}$ super-cooled liquid droplets as well as ice particles can exist; however, clouds containing both kinds of hydrometeors are called mixed phase clouds. At lower temperatures ($T < 235\,\mathrm{K}$), only the solid phase exists, i.e. we find pure ice clouds.

Clouds influence the hydrological cycle, e.g. by forming precipitation. Further clouds influence the general circulation by diabatic processes induced by phase changes (Reed et al., 1992; Wernli and Gray, 2024). Clouds also affect the system's energy budget by interaction with solar and thermal radiation. Incoming solar radiation is partly scattered and reflected back to space (albedo effect) whereas the Earth's outgoing infrared radiation is partly absorbed and re-emitted at a different temperature (greenhouse effect). The IPCC report 2021 states that the overall cloud feedback on climate is positive, hence enhancing global

warming. However, there are still many cloud regimes with large uncertainties regarding their impact on the global energy budget (Forster et al., 2021). One of these cloud regimes are high-altitude (i.e. cold temperature regime), pure ice clouds, referred to as cirrus. Since the albedo and greenhouse effect of cirrus are of the same magnitude but different sign, the combined effect can lead to a net cooling or net warming depending on microphysical properties (see, e.g., Zhang et al., 1999; Fusina et al., 2007; Joos et al., 2014). Overall, our understanding of clouds and the relevant and dominant processes is still quite limited,

especially for pure ice clouds (Krämer et al., 2020).

For pure ice clouds at low temperatures, the pathway of formation remains unclear a priori. The different formation mechanisms were recently summarized and classified in the following ways (see, e.g., Krämer et al., 2016; Wernli et al., 2016):

- Formation pathways related to freezing of pre-existing cloud droplets (i.e. in the mixed-phase temperature regime and close to water saturation) are termed liquid origin formation.

- Formation pathways related to direct formation of ice crystals from vapor at (liquid or solid) aerosols at low temperatures and below water saturation are termed in-situ formation.

It is quite difficult to determine the formation pathway of ice crystals from observations. For measured ice particles with imaging techniques there are some hints that complex shapes and large particle sizes are most probable for liquid origin ice crystals, whereas in-situ formed ice crystals remain small, and their shapes remain simple, e.g., as quasi-spherical shapes (Wolf

et al., 2018). This can be explained by the amount of available water vapor, which can be used for diffusional growth. At the mixed-phase regime, more water vapor is available leading to larger and more complex shapes, whereas at low temperatures the amount of water vapor is very limited leading to small sized and simple shaped ice crystals. It also seems to be probable that liquid origin ice clouds consist of much more ice particles than in-situ formed ice clouds (Krämer et al., 2016, 2020). This can be explained by the formation process. Liquid origin ice particles stem from pre-existing cloud droplets; the number

concentration of water droplets in liquid clouds is usually in the order of some hundreds of particles per cubic centimeter and thus much larger than the amount of available ice nuclei at low temperatures.

There are some indications that we might be able to use the particle's properties themselves for a classification of the formation pathways (see, e.g., Luebke et al., 2016; Wolf et al., 2018). However, such evaluations are just based on single cases of measurements together with trajectory calculations and have an inherit uncertainty, which cannot be quantified. Another approach

to identify the ice formation pathway is to employ retrievals from remote sensing to calculate cloud properties and then to search for prescribed characteristics (see Urbanek et al., 2017; Huo et al., 2020).

The formation pathway can also be estimated by using model simulations. Gasparini and Lohmann (2016) utilized a temperature based criterion to classify cirrus cloud types. Wernli et al. (2016) utilized Lagrangian trajectories to identify the origin

of the cirrus cloud. If cirrus originated from the liquid phase (via a mixed phase cloud) or from clear sky, the resulting ice cloud was classified as *liquid origin* or *in-situ*, respectively. However, both approaches do not assess the formation pathway directly and do not consider transport relative to the air parcel, i.e. neither sedimentation nor turbulent mixing. In addition, the evaluations were carried out on coarse grid resolutions (climate model or reanalysis data)

A complementary viewpoint from the (even small scale) modeling perspective is necessary in order to give a more rigorous insight into the different formation mechanism in cloud systems, especially for vertically extending systems such as convective clouds or warm conveyor belts that show both the mixed-phase and pure ice thermodynamic regime.

A particle-by-particle direct numerical simulation of each cloud particle is not feasible for realistic clouds (Morrison et al., 2020). Instead, bulk schemes are a commonly used model approach to parameterize cloud physics in large scale atmospheric models with spatial resolutions of several kilometers. They also have been utilized in high resolution large eddy simulations and, on the other end of the spatial and temporal scale, in climate models. Instead of describing the evolution of individual cloud particles, bulk schemes describe the evolution of mean quantities of an ensemble of particles. The cloud particles are sorted into preset classes and assigned a (semi-empirical) type of size distribution. Integrating the evolution equation over the size distribution we obtain the temporal evolution of general moments (Hulburt and Katz, 1964). Instead of size distributions, often the particle mass is used as a variable of distributions, thus leading to general moments of mass; in this work, we use this approach, which is more physical since mass is the conserved quantity (see, e.g., Seifert and Beheng, 2006). Bulk schemes were first introduced by Kessler (1969), but only considered cloud droplets and raindrops. Since the work of Lin et al. (1983) ice particles are also commonly included in bulk schemes. For practical reasons, we use a finite number of moments in order to describe the system. The order of the bulk scheme is set by the number of predicted moments. One-moment bulk schemes typically predict the mass content (first moment) of a cloud particle class. Two-moment schemes predict the number concentration (zeroth moment) in addition to the first moment. For a meaningful closure of the system, we usually fix the type of the underlying mass distribution.

Some common choices for bulk schemes in numerical weather prediction and atmospheric research models are: Seifert and Beheng (2006) for the ICON model (Zängl et al., 2015; Heinze et al., 2017), Thompson et al. (2008) and Morrison et al. (2009) for the WRF model (Skamarock et al., 2019), Field et al. (2023) for the Unified model (Walters et al., 2019), and the bulk scheme employed in the IFS model (ECMWF, 2023), respectively. The predicted particle properties (P3) scheme (Morrison and Milbrandt, 2015) is also commonly employed in cloud research. Instead of multiple distinct ice particle classes (cloud ice, snow, graupel, hail) they employ free ice-phase classes that can evolve into any ice particle type (or mixture thereof).

In this paper we introduce a novel bulk scheme, the ice modes scheme. This new scheme is based on the standard two-moment scheme in ICON (Seifert and Beheng, 2006) (SB hereafter). However, instead of a single ice crystal (or 'cloud ice') class, it features several classes, each equipped with their unique formation process. Apart from the unique formation processes these ice modes use the same parameterisations for shape, terminal velocity and all other microphysical processes. They are subject to advection, turbulent mixing and sedimentation. Ice modes can also (like all microphysics classes) coexist, i.e. like in the original SB scheme, multiple ice formation processes can still be active at the same time if the relevant thermodynamic requirements are met.

The purpose of the new ice modes scheme is not to perform better than the base scheme of SB concerning quantifiable metrics like precipitation rates and patterns, or even scores of operational numerical weather prediction models. In that regard the aim is that the ice modes schemes performs close to and ideally the same as the original SB scheme. The benefit of employing multiple cloud ice classes with unique formation processes is that the ice modes schemes retains the information of how cloud ice originally formed. This allows a novel way of addressing research questions for glaciated clouds. Liquid origin vs in-situ formation can be quantified by the comparing the amount of cloud ice stemming from frozen droplets and directly from the vapor phase formed ice. Relevance of secondary ice processes can be assessed as mass content and number concentrations of primary and secondary ice are separate tracers. Also note that the separation into these ice modes itself is not artificial. Conceptually we could attribute a distinct formation process to each ice crystal measured in real cloud or laboratory experiments as it is a unique property assigned to the particle at its formation.

Hence the new ice mode schemes is a useful tool for the investigation of cloud evolution. Additionally it can evaluate the sensitivity of different ice formation pathways to environmental conditions e.g. thermodynamic variables and domain properties like spatial and temporal resolution. It is also useful to test new ice nucleation parametrisations as the individual impact of each routine on the model cloud can be directly assessed.

There are two other approaches to investigate ice formation pathways in models. First is to track (usually temporally or spatially accumulated) process rates of ice nucleation as additional model output variables. This approach has been used in studies that evaluate the impact of individual cloud processes including nucleation (see, e.g., Barrett and Hoose, 2023; Han et al., 2023; Oertel et al., 2023; Han et al., 2024; Schwenk and Miltenberger, 2024). However, this approach only provides the amount of ice that formed from each individual formation process. It does not track what occurs to this ice after formation as the signal gets blurred by advection, turbulent mixing, sedimentation and removal by collisions, melting and evaporation, respectively. Hence, this approach provides how much ice was formed by each process and the ice mode schemes provides information on how much ice from each formation process is left. These are not derivative but complementary viewpoints.

One other approach that can (in theory) track all microphysical processes including formation and retain that information are (Lagrangian) particle models. These models simulate the properties of 'super particles' that represent an ensemble of usually $10^1$ to $10^4$ members. These particles move freely through the model domain in a Lagrangian sense and in general do not belong to a certain preset class but instead are able to evolve dynamically depending on environmental conditions and their interaction with other particles. There are only few particle models that include the ice phase: Sölch and Kärcher (2010), Unterstrasser and Sölch (2010), Brdar and Seifert (2018), Shima et al. (2020). However, there is no particle model yet that (directly) models all ice formation processes relevant for both, the liquid origin and the in-situ formation pathway, respectively. Another issue is that particle models are computationally expensive (Morrison et al., 2020). Hence, they are only employed in parcel models or high-resolution (cloud-resolving) models to investigate atmospheric systems on the micro to meso-scale. The ice modes bulk scheme introduced in this work can be used to investigate synoptic-scale systems like Warm Conveyor Belts and even be used in global simulations.

In a first application of the new model we investigate the formation of ice clouds for an idealized test case of convection, and determine the formation pathways of (anvil) ice clouds. Although it was always assumed that the anvil of a convective thunder-

storm should be mainly formed by frozen cloud droplets transported upwards, it was not shown before (Gasparini et al., 2018).
Using our newly developed ice modes scheme, we can investigate this statement quantitatively.

In Section 2 we describe the model equations and relevant microphysical parametrisations affecting ice in the model and than present a idealised simulation of a convective cloud in Section 4. We compare our results with a reference simulation that uses the standard SB scheme and also demonstrate the capabilities of the new scheme to characterize the origin of cloud ice and impact of different nucleation parametrisations.

## 2  Model description

The basis for our model development is the two-moment scheme of SB. The original SB scheme distinguished between six classes of cloud particles: cloud droplets, rain, (cloud) ice, snow, graupel and hail. While the formulation for the liquid droplet's treatment in the model remains unchanged, we reformulate the ice microphysics, and also the interaction of ice and water particles in terms of collision processes. In the following the new scheme is described.

### 2.1  General settings

The model formulation relies on the usual approach of a bulk model with two (general) moments. Instead of computing the temporal and spatial evolution of a (maybe multivariate) mass distribution with a Boltzmann-type evolution equation, we use the integrated version, i.e., the evolution equations for moments of the underlying mass distribution of the respective class of hydrometeors. The moments are defined in the usual way, i.e.

$$M_i^k := \int_0^\infty x^k f_i(x)\mathrm{d}x \tag{1}$$

with the mass of particles $x$ in kg and the mass distribution $f_i$ in $\mathrm{kg}^{-1}$ for the respective class $i$ of hydrometeors. The mass distribution describes the number is normalized by the total number concentration of particles $n_i$. For a meaningful closure of the systems of equations, we have to choose a suitable type of mass distribution. We generally assume that the particle mass distributions can be represented by generalized Gamma-distributions (see also Seifert and Beheng, 2006) of the form

$$f_i(x) = A_i x^{\nu_i} \exp(-\lambda_i x^{\mu_i}) \tag{2}$$

where the shape parameters $\nu_i$ and $\mu_i$ are prescribed, $\lambda_i$ and $A_i$ are linked to the zeroth and first distribution moments, the number concentration $n_i$ and the mass content $q_i$

$$n_i = M_i^0 = \int_0^\infty f(x)\,\mathrm{d}x = \frac{A}{\mu \lambda^{\frac{\nu+1}{\mu}}} \Gamma\left(\frac{\nu+1}{\mu}\right) \tag{3}$$

$$q_i = M_i^1 = \int_0^\infty f(x)x\,\mathrm{d}x = \frac{A}{\mu \lambda^{\frac{\nu+2}{\mu}}} \Gamma\left(\frac{\nu+2}{\mu}\right) \tag{4}$$

See Appendix B for details on the properties of the gamma distribution and the analytical solution of the integrals.

For the formulation of single particle processes (e.g. growth or sedimentation), we have to introduce size-mass relations and velocity-mass relations of the form

$$D_i(x) = a_i x^{b_i}, \quad v_i(x) = \alpha_i x^{\beta_i} \tag{5}$$

with the (generalized) size $D_i$ and the terminal velocity $v_i$ for a particle of mass $x$ within a class $i$, respectively. These relations represent the different shapes of particles from different classes. The remaining classes of hydrometeors are labeled by indices, i.e. $c$ for cloud droplets, $r$ for rain drops, $s$ for snow, $g$ for graupel, and $h$ for hail, respectively. The old class cloud ice is now splitted into five classes. For details of the formulation and the determination of the shapes for the used hydrometeor classes we refer to Seifert and Beheng (2006).

For our newly introduced ice modes we use the same parameters of the distribution (2) and the relations (5) for all classes. While the formation pathway of an ice crystal might have an impact on its shape, the morphology of the particle is mainly determined by the environmental conditions encountered during its growth, e.g., temperature and humidity (see, e.g., Magono and Lee, 1966; Kobayashi, 1967; Libbrecht, 2005; Pruppacher et al., 2010), which is not accounted for in the SB scheme. Using the same coefficients and parametrisations for each ice mode apart from its source also has the advantage that it allows a more concise interpretation on the impact of the individual ice formation pathways on the cloud evolution.

The time evolution of the kth-moment $M_i^k$ of an ice mode $i$ (or another class of hydrometeors) is governed by

$$\frac{\partial M_i^k}{\partial t} + \boldsymbol{\nabla} \cdot \left[ \boldsymbol{v} M_i^k \right] + \frac{\partial \bar{v}_{i,k} M_i^k}{\partial z} - \boldsymbol{\nabla} \cdot \left[ K_h \boldsymbol{\nabla} M_i^k \right] = \text{Source/Sink} \tag{6}$$

The terms on the left-hand side describe the effects of advection with the mean wind velocity $\boldsymbol{v}$, sedimentation with the weighted mean fall-speed $\bar{v}_{i,k}$ and turbulent mixing with the mean turbulent diffusivity of heat $K_h$, respectively. On the right-hand side are the source and sink terms for the particle formation, which is unique for each ice mode, deposition of water vapor and evaporation and a number of collisions processes, most importantly aggregation and riming, respectively. These equations must be coupled with a model for atmospheric flows, as, e.g., suitable approximations of the Navier-Stokes equations within the ICON model.

In the following we will focus on the description of the ice related physics and refer to Seifert and Beheng (2006) for an in-depth description of the warm- and mixed-phase microphysics.

## 2.2 Ice formation pathways

For the treatment of ice particles, we introduce new classes of ice particles, discriminated by their formation mechanisms. The ice modes scheme features five independent ice classes instead of a single one (former class "cloud ice"), called ice modes in this work, each with their unique formation pathway. For each new ice mode, we introduce number concentrations and mass content, respectively. The other classes (graupel, hail and snow) remain unchanged; however, we have to reformulate the processes of interactions between the former class "cloud ice" and other variables (e.g. for collision processes, see below). The new classes are as follows:

- Homogeneous freezing of cloud droplets $\qquad n_{frz}, q_{frz}$
- Immersion freezing of cloud and rain droplets $\qquad n_{imm}, q_{imm}$
- Freezing of solution droplets (homogeneous nucleation) $\qquad n_{hom}, q_{hom}$
- Deposition nucleation $\qquad n_{dep}, q_{dep}$
- Secondary ice from rime splintering $\qquad n_{sec}, q_{sec}$

The sum of all ice modes represent all ice crystals present ($n_{tot}, q_{tot}$).

### 2.2.1 Homogeneous freezing of cloud droplets

Cloud droplets are considered to be pure water droplets, thus freezing homogeneously at temperatures below the triple point $T_m$ but at water saturation. Homogeneous freezing of cloud droplets is the source for the FRZ ice mode with the bulk quantities $q_{frz}$ and $n_{frz}$. The stochastic process of homogeneous freezing is described by a nucleation rate $J_{hom}$ in $\mathrm{m^{-3}s^{-1}}$, which depends on temperature only (since existence of cloud droplets requires water saturation), thus the change in number and mass concentrations of the ice mode can be described by

$$\frac{\partial n_{frz}}{\partial t} = J_{hom}q_c, \quad \frac{\partial q_{frz}}{\partial t} = J_{hom}q_c x_c \frac{\nu_c + 2}{\nu_c + 1} \tag{7}$$

using the mass content of cloud droplets $q_c$, the mean mass of cloud droplets $x_c$ and the distribution shape parameter $\nu_c$ of the cloud mass distribution (generalized gamma distribution), respectively. For the formulation of the homogeneous freezing coefficient of cloud droplets in $\mathrm{\mu kg^{-1}s^{-1}}$, we use the fit of Cotton and Field (2002) to the formulation of Jeffery and Austin (1997):

$$\log\left(\frac{J_{hom}}{\rho_w}\right) = \begin{cases} -243.4 - 14.75T - 0.307T^2 - 0.00287T^3 - 1.02 \times 10^{-5}T^4 & \text{for} \quad T \le 243\,\mathrm{K} \\ -7.63 - 2.996(T + 30) & \text{for} \quad T > 243\,\mathrm{K} \end{cases} \tag{8}$$

whereas $\rho_w$ denotes the liquid water mass density. The parametrisation for the nucleation rate is only valid at water saturation. However, due to saturation adjustment (see SB), the system stays at water saturation permanently as long as cloud droplets are present.

Homogeneous freezing of raindrops in the SB scheme is omitted, since raindrops freeze rapidly by heterogeneous freezing before reaching temperature levels close to the homogeneous freezing temperature.

### 2.2.2 Homogeneous freezing of solution droplets

Aqueous solution droplets, i.e. liquid aerosol particles can be supercooled to lower temperatures than pure liquid droplets. The solute obviously impedes the establishment of a critical cluster. Koop et al. (2000) showed that the effect of solutes on the freezing temperature is driven by their thermodynamic (equilibrium) quantities, which can be expressed in terms of water activity, that is defined as the water saturation vapor pressure ratio between the solution and pure water $a_w = e_{sol}/e_{liq}$. When we assume that the solution droplet is in a thermodynamic equilibrium with its environment than the freezing temperature is

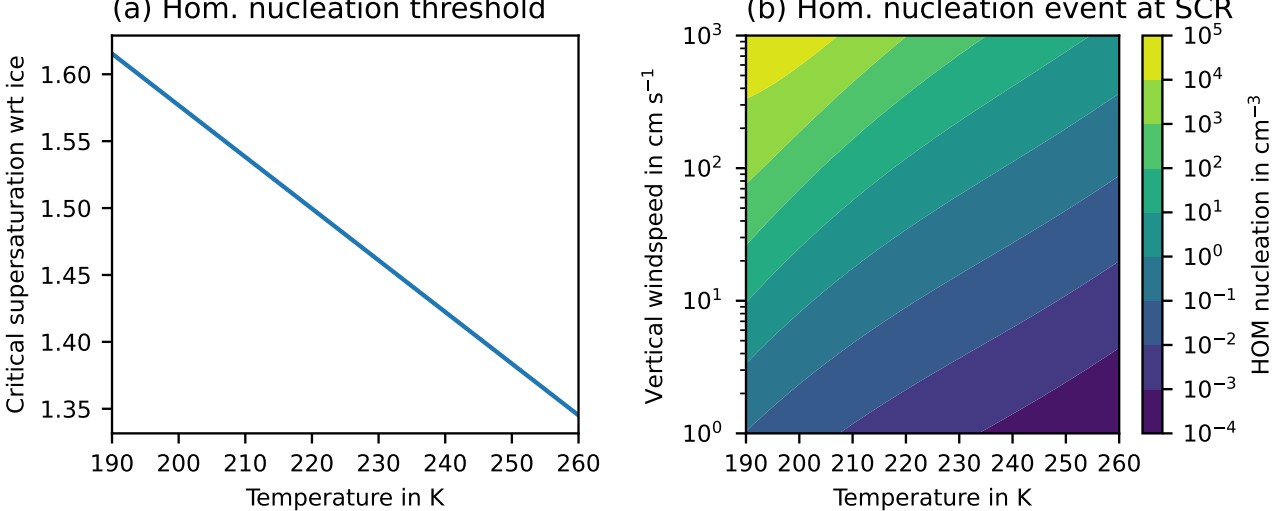

**Figure 1.** Homogeneous nucleation following Kärcher et al. (2006) for no pre-existing ice. (a) Critical supersaturation $S_{cr}$ serves as a nucleation threshold and (b) number of ice crystals after a homogeneous nucleation event at $S_{cr}$.

independent of the choice of chemical droplet composition, at least for inorganic compounds. This translates into a nucleation rate for solution droplets, which solely depends on the environmental conditions, i.e. temperature and supersaturation with respect to ice, but not on the substance in the water drops. The nucleation rate can be formulated using a threshold of critical
supersaturation, as could be shown by Spichtinger et al. (2023), e.g., using the formulation by Ren and MacKenzie (2005)

$$S_{cr} = 2.349 - T/259. \tag{9}$$

Panel (a) in Figure 1 shows that quite high supersaturations ($S_i > 1.4$) are necessary for homogeneous nucleation events to occur, especially in low temperature regimes ($T < 230\,\mathrm{K}$) where freezing of pure water droplets is no longer possible and homogeneous nucleation commonly observed. This ice formation process is called homogeneous nucleation and the source
term for the HOM ice mode with the bulk properties $n_{hom}$ and $q_{hom}$. Generally, we would obtain similar equations for the change in $n_{hom}, q_{hom}$ as for the freezing of cloud droplets (see Section 2.2.1). However, since the system is in a non-equilibrium state and there is no boundary condition (like the assumption of water saturation as above), we would have to represent the evolution of the saturation ratio, changed also by diffusional growth; this would require a small timestep for the numerical scheme and thus is not feasible.
Therefore, we use the parameterisation by Kärcher and Lohmann (2002), which describes the homogeneous nucleation event in an ascending air parcel. When an air parcel ascends adiabatically supersaturation is generated by adiabatic cooling. When the critical supersaturation $S_{cr}$ is reached, homogeneous nucleation is triggered. The newly nucleated ice crystals deplete supersaturation by depositional growth. The competition between generating supersaturation by adiabatic cooling and depleting supersaturation by depositional growth in an adiabatically ascending air parcel, driven by a constant wind velocity $w$ can be

described as

$$\frac{dS}{dt} = a_1 S w - (a_2 + a_3 S) R_i(t) \tag{10}$$

with the parameters

$$a_1 = \frac{L_s M_w g}{c_p R T^2} - \frac{M_a g}{RT}, \ a_2 = \frac{T k_b}{e_{sat,i}}, \ a_3 = \frac{L_s^2 M_w m_w}{c_p p T M_a} \tag{11}$$

$a_1$ describes the effect of adiabatic cooling, where $a_2$, $a_3$ and $R_i$ describe the depletion of supersaturation by depositional growth as a function of time. The parameters and process of depositional growth will be further explained in Section 2.3.

If the air parcels continues to rise, aqueous solution droplets continue too freeze. When the newly nucleated ice crystals deplete more supersaturation than is being generated by adiabatic cooling, the maximum value of the supersaturation $S_*$ is reached. The supersaturation is depleted and once it reaches $S_{cr}$ the freezing of solution droplets stops. The integrated number of solution droplets frozen within the freezing time interval $t_*$ described the number of ice crystals from the homogeneous nucleation event. Numerical simulations suggest that the freezing time interval $t_*$ is short and the maximum value of supersaturation $S_*$ is approximately equal to $S_{cr}$ (Kärcher et al., 2006). Thus the number concentration of ice particles in a homogeneous nucleation event can than be estimated as

$$n_{hom} = w \frac{a_1 S_{cr}}{a_2 + a_3 S_{cr}} \frac{1}{R_{im}(r_0)} \tag{12}$$

$R_{im}$ is the analytical approximation of the integral that describes the growth of the solution droplet within the freezing time interval where we refer to Kärcher et al. (2006) for a detailed derivation. $r_0 = 27.2\,\mathrm{nm}$ is the radius of the monodisperse aqueous solution droplets. We obtain the bulk mass of the newly nucleated homogeneous ice mode as

$$q_{hom} = \frac{4}{3}\pi \rho_i \left( \frac{r_0(1 + b_{KH} r_0) - 1}{b_{KH}} \right)^3 n_{hom} \tag{13}$$

with the parameter $b_{KH} = \frac{\alpha_d v_{th}}{4 D_v}$. $\alpha_d$ is the deposition coefficient, $v_{th}$ is the mean molecular velocity of water vapor, and $D_v$ denotes the diffusivity.

Panel (b) in Figure 1 shows the number concentration of ice $n_i$ after a homogeneous nucleation event at critical supersaturation $S_{cr}$ for vertical velocities ranging from synoptic velocities (up to $10\,\mathrm{cm\,s^{-1}}$), gravity waves (up to $100\,\mathrm{cm\,s^{-1}}$) and convection (up to $1000\,\mathrm{cm\,s^{-1}}$). Homogeneous nucleation is strongly sensitive to the model level vertical velocity and large nucleation events can only be represented if vertical velocity are resolved, and no other nucleation mechanisms disturbs the effect of homogeneous nucleation. Nucleation events at synoptic vertical velocity will be more common in the model and nucleate up to $1\,\mathrm{cm\,s^{-1}}$ ice crystals. Note that the parametrisation of Kärcher et al. (2006) does not explicitly scale with the microphysics time step as it describes an entire nucleation event and not a nucleation rate. Shorter time steps might lead to higher number concentrations because newly nucleated ice crystals have less time to deplete the supersaturation until a new event is allowed to trigger. However, the homogeneous ice mass content is still constrained by the availability of supersaturation, i.e. by thermodynamics.

The ice modes scheme uses the extension of Kärcher et al. (2006) where the effect of pre-existing ice depleting supersaturation is taken into account as an equivalent to a fictitious downdraft velocity $w_{pre}$

$$w' = w - w_{pre} = w - \frac{a_2 + a_3 S_i}{a_1 S_i} R_{i,pre} \tag{14}$$

with $w$ as the model level vertical velocity and $R_{i,pre}$ as the mean radius of the pre-existing ice. Pre-existing ice usually originates from a heterogeneous ice mode or a prior homogeneous nucleation event, as we will show in the simulation results.

**2.2.3 Heterogeneous nucleation**

In the new scheme we use the two nucleation pathways, i.e. immersion freezing and deposition nucleation. Both ice formation pathways depend on temperature, supersaturation and properties of the INP like effectiveness and number of sites causing nucleation per unit surface area (Vali et al., 2015). Both immersion freezing of raindrops and cloud droplets are considered in this model.

We obtain the number concentration for the immersion freezing ice mode as the sum of freezing cloud and rain droplets as

$$\frac{\partial n_{imm}}{\partial t} = C_{IMM}|_c - n_{inact} + \frac{\partial n_{imm}}{\partial t}|_r \tag{15}$$

and the mass content as

$$\frac{\partial q_{imm}}{\partial t} = \frac{\partial n_{imm}}{\partial t}|_c x_c + \frac{\partial q_{imm}}{\partial t}|_r \tag{16}$$

with the mean cloud droplet mass $x_c$. Similarly we obtain for the number concentration of the deposition nucleation ice mode

$$\frac{\partial n_{dep}}{\partial t} = C_{DEP} - n_{inact} \tag{17}$$

and the mass content

$$\frac{\partial q_{dep}}{\partial t} = \frac{\partial n_{dep}}{\partial t} x_{i,min} \tag{18}$$

with the minimum mass of ice crystals in the model $x_{i,min}$. $C_{IMM}$ and $C_{DEP}$ describe the number concentration in $\mathrm{m}^{-3}$ of activated INPs for immersion freezing (of cloud droplets) and deposition nucleation, respectively. The model does not

feature an explicit model for INP or aerosols in general. Instead the number of activated INPs $n_{inact}$ in $\mathrm{m}^{-3}$ is tracked as an additional tracer and advected along with the other ice bulk properties. In further nucleation events $n_{inact}$ is subtracted from the INP number. $n_{inact}$ relaxes back to zero in a ice free environment

$$\frac{\partial n_{inact}}{\partial t} = \begin{cases} -\frac{n_{inact}}{\tau_{inact}} & \text{if } q_{tot} = \frac{\partial n_{imm}}{\partial t} = \frac{\partial n_{dep}}{\partial t} = 0 \\ \frac{\partial n_{imm}}{\partial t} + \frac{\partial n_{dep}}{\partial t} & \text{else} \end{cases} \tag{19}$$

with the relaxation time scale $\tau_{inact} = 600\,\mathrm{s}$ and the total ice content (as the sum of all ice modes) $q_{tot}$.

First, we describe immersion freezing for rain droplets, i.e. for large water droplets. Bigg (1953) performed laboratory studies

investigating the freezing of supercooled purified water in the presence of INP and estimated the probability of a raindrop freezing depending on temperature and droplet volume. Using this results the freezing rate of rain can be expressed as

$$\frac{\partial n_{imm}}{\partial t}|_r = -J_{bigg}q_r = -A_{imm}\rho_w^{-1}\exp(B_{imm}(T_m - T) - 1)q_r \tag{20}$$

where we use the coefficients $A_{imm} = 200\,\mathrm{m^{-3}s^{-1}}$ and $B_{imm} = 0.65\,\mathrm{K^{-1}}$ for rain water due to Barklie and Gokhale (1959). We obtain the mass of frozen raindrops as

$$\frac{\partial q_{imm}}{\partial t}|_r = -A_{imm}\rho_w^{-1}\exp(B_{imm}(T_m - T) - 1)q_r\overline{x}_r Z_r \tag{21}$$

with the second moment of the rain mass distribution $Z_r = M_r^2$ and the melting temperature $T_m$. The mass and number concentration of frozen raindrops is partitioned into the immersion freezing ice mode (IMM), graupel and hail depending on its diameter. Raindrops smaller than $0.5\,\mathrm{mm}$ freeze into ice, raindrops with sizes between $0.5\,\mathrm{mm}$ and $1.25\,\mathrm{mm}$ are shifted into the graupel class, and raindrops larger than $1.25\,\mathrm{mm}$ are considered as hail.

Second, we present the nucleation schemes for immersion freezing of cloud droplets and deposition nucleation, respectively, resulting into changes in the ice modes $n_{imm}, q_{imm}$ (immersion freezing) and $n_{dep}, q_{dep}$ (deposition nucleation). The ice mode scheme offers the choice between the three heterogeneous nucleation schemes, i.e. Hande et al. (2015), Ullrich et al. (2017), Phillips et al. (2008). In the following, we will introduce all three options. The choice of heterogeneous nucleation schemes has a large impact on ice formation. We will compare the impact of the heterogeneous nucleation choice on the ice modes for an idealized convective case in Section 4.3.

Hande et al. (2015) (HA15) considered dust as the main source for INPs over Europe. They used the COnsortium for Small-scale MOdelling (COSMO) meteorological model coupled to the MUlti-Scale Chemistry Aerosol Transport (MUSCAT) to simulate Sahara dust outbreaks for the year 2008. From the statistics of simulated dust concentrations HA15 calculated atmospheric profiles of potential INPs. Since Sahara dust outbreaks shows a strong seasonal variability HA15 provided mean profiles for each season. The number of active INPs for immersion freezing was than parameterised using the laboratory results of Niemand et al. (2012) for dust particles

$$C_{IMM}(T) = A_H \cdot \exp\left[-B_H(T - T_{H,min})^{C_H}\right] \tag{22}$$

valid for the temperature range from $T_{H,min} = 237.15\,\mathrm{K}$ to $261.15\,\mathrm{K}$ with the set of coefficients $A_H, B_H, C_H, T_{H,min}$ being chosen depending on the season (see Table 1 in Hande et al. (2015)). The number concentration of active INPs for deposition nucleation was estimated using the parametrisation of Steinke et al. (2015)

$$C_{DEP}(T, S_i) = C_{IMM}(T) \cdot (a_H \arctan(b_H(S_i - 1) + c_H) + d_H) \tag{23}$$

for temperatures between $T_{H,min} = 220\,\mathrm{K}$ and $253\,\mathrm{K}$. Again, the coefficients depend on the season (see Table 1 in Hande et al. (2015)).

Ullrich et al. (2017) (UL17) used 11 years of data from ice nucleation experiments in the Aerosol Interaction and Dynamics in the Atmosphere (AIDA) cloud chamber to develop an empirical parametrisation for both immersion freezing and deposition

nucleation. Dust and soot samples of different types, collected in various locations of the world were analysed in AIDA. The data set includes the results published in Niemand et al. (2012), which where also used in the HA15 parametrisation. However, their are some differences in their approach to calculate the density of active surfaces sites (see UL17) resulting in a light shift to higher densities. In this study we only consider the dust mode. We obtain the number of active INP for immersion freezing and deposition nucleation, respectively, for dust by

$$C_{IMM/DEP}(T, S_i) = n_a\big(1 - \exp\big[-n_{s,IMM/DEP}(T, S_i)SA_a\big]\big) \tag{24}$$

with the aerosol number concentration $n_a$ in $\text{m}^{-3}$, the aerosol surface area concentration $SA_a$ and the ice nucleating active surface site density $n_{s,IMM/DEP}$ in $\text{m}^{-2}$ for immersion freezing and deposition nucleation, respectively. The ice nucleating active surface site density for immersion freezing follows a simple exponential temperature profile

$$n_{s,IMM} = \exp[150.577 - 0.517T] \tag{25}$$

and for deposition nucleation

$$n_{s,DEP} = \exp\Big[\alpha(S_i - 1)^{1/4}\cos(\beta(T - \gamma))^2 \operatorname{arccot}(\kappa(T - \lambda))\Big] \tag{26}$$

where the fixed coefficients $\alpha$, $\beta$, $\gamma$, $\kappa$ and $\lambda$ can be found in UL17. For dust size distribution we use a sum of three lognormal distribution modes. The distribution parameters were chosen such that the immersion freezing temperature profile for UL17 is similar to HA15.

Phillips et al. (2008) (PH08) developed an empirically derived framework for heterogeneous nucleation of multiple aerosol species.

INP are grouped into three basic types: dust/metallic aerosols, inorganic black carbon and insoluble biological aerosol like bacteria and pollen. The basic assumption is that the ice nucleating activity of insoluble aerosol depends on its number of sites causing nucleation (active sites) and hence is proportional to the total aerosol surface area. The number concentration of active INP for a aerosol group X was parameterised as

$$C_{INP,X} = \int_{D_{X,min}}^{\infty} (1 - \exp[-\mu_X(D_X, T, S_i)])f_X(D_X)\,\mathrm{d}D_X \tag{27}$$

where $\mu_X$ is the Poisson distributed average of the number of activated INPs for each aerosol particle of size $D_X$. $\mu_X$ was empirically determined by using observational and laboratory data (see PH08). The ice mode schemes uses a simple lookup table and 2D interpolation to determine the fraction of activated INPs as function of temperature and supersaturation for each aerosol type. Lognormal size distributions are assumed for all aerosol types. The number of INPs is the sum over all three aerosol types with the associated number concentration $n_X$ and active fraction $C_{INP,X}$

$$C_{INP} = \sum_X C_{INP,X}(T, S_i)n_X \tag{28}$$

where we obtain the number of INPs for immersion freezing at and for deposition nucleation below saturation wrt to water. The initial number concentration of soot particles and biological aerosol is fixed and the number concentration of dust varies with altitude $z$ following a prescribed background profile

$$n_{dust}(z) = n_{dust,0} \exp\left[\frac{z_{dust,0} - z}{z_{dust,e}}\right] \tag{29}$$

Unless otherwise noted we only consider the dust mode for PH08 in this work. We chose $z_{dust,0}$ such that the maximum number of INPs activated for immersion freezing is the same as for HA15.

Panel (a) in Figure 2 shows the temperature profiles for immersion freezing parametrisations (IMM) and contour plots of deposition nucleation (c-d). PH08 uses geometric altitude as a coordinate for dust concentration. We transform the altitude into a temperature coordinate using the ICAO Standard Atmosphere. The profile labeled PH08+ considers also soot and biological aerosols in addition to dust.

The exponential profiles of immersion freezing for HA15 and UL17 are very similar starting both close to $260\,\mathrm{K}$ and being capped at the homogeneous freezing temperature threshold. It is not surprising that we only observe minor differences since active INPs for immersion freezing in both parametrisations is based on mostly the same laboratory datasets. On the other hand immersion freezing for PH08 starts at higher temperatures but increases less steeply past $260\,\mathrm{K}$. Especially for convectively driven clouds, the earlier onset of immersion freezing has potentially a strong impact on the evolution of the ice phase. PH08+ shows activation of INP at even higher temperature caused by soot and biological aerosol. However, since we did not tune this version of the PH08 scheme to HA15 the maximum number of active INP is lower.

### 2.2.4 Secondary ice production

The only secondary ice process considered in this study is rime splintering (RS). Ice crystals stemming from secondary ice processes are referred to as secondary ice particles (SIP). Supercooled droplets colliding with ice particles (riming), especially graupel and hail, can throw off small ice splinters, which can grow into SIP. This phenomenon was first investigated by Macklin (1960). Various physical mechanism have since been suggested: Macklin (1960) reported that riming caused the growth of fine ice structures that would break to create SIP while Choularton et al. (1978) suggested that shattering of freezing droplets produces SIP. Hallett and Mossop (1974) counted ice crystals in a light-beam beneath a metal rod, which moved through the cloud chamber and swept up water droplets causing riming. By comparing the enhanced number of ice crystals to the background number of the cloud chamber they derived a profile depending on cloud temperature. The standard parametrisation for RS is based on this study and often referred to as the Hallett-Mossop process.

Emersic and Connolly (2017) investigated riming events using high-speed cameras concluded that even small ice crystals and not only large rimers like graupel could potentially produce a significant amount of SIP. Seidel et al. (2024) investigated rime splintering with high-speed video microscopy, IR thermography and a custom-build ice counter. They could not reproduce the results of Hallett and Mossop (1974). In general they found only insignificant amounts of SIP production during riming, which can not explain the amount of SIP expected in convective and frontal clouds.

Overall Korolev and Leisner (2020) found no consistency in measured or estimated rime splintering rates between various

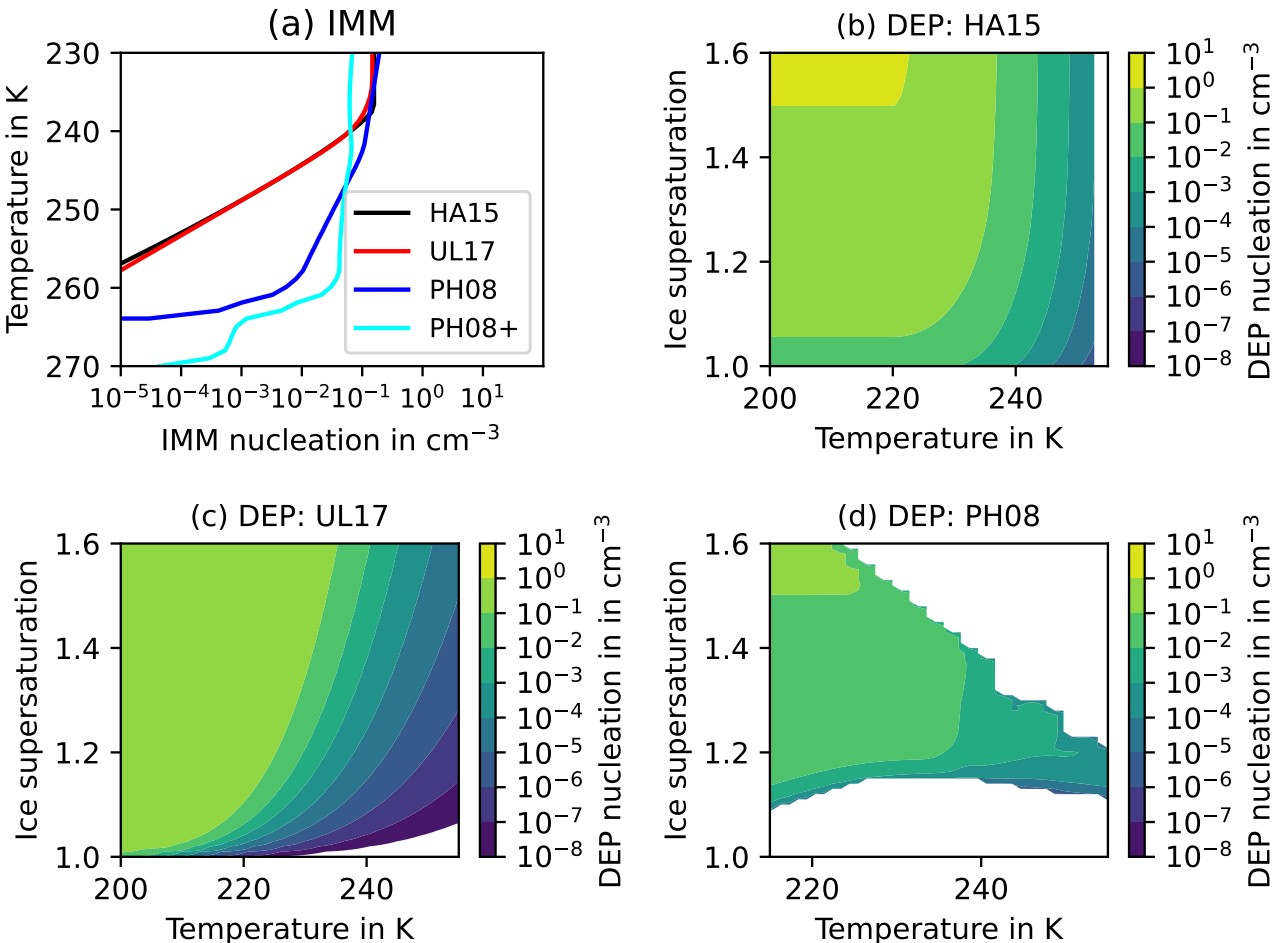

**Figure 2.** Number concentration of activated INPs for (a) Immersion freezing with profiles for HA15, UL17, PH08 and PH08+ schemes and (b-d) deposition nucleation as a function of $T$ and $S_i$ for (b) HA15, (c) UL17, (d) PH08.

groups and attributed the discrepancy to different laboratory setups and techniques. Despite RS being the most commonly included secondary ice mechanism in numerical cloud models, the physical understanding is severely lacking and thus the development of a physically based parameterisation seems unfeasible at the moment (Korolev and Leisner, 2020). Still we will use the parameterisation of RS based on the dataset of Hallett-Mossop in this study as it is part of the standard SB scheme.

In Hallett and Mossop (1974) rime splintering occurred within a narrow temperature range of $T_{rs,min} = 265\,\text{K}$ and $T_{rs,max} =$

270 K. A fit onto their dataset is used as a function of temperature and rimed mass $q_{rime}$ in kg as it is implemented in SB

$$\frac{\partial n_{sec}}{\partial t} = q_{rime} C_{RS} \frac{T - T_{rs,min}}{T_{rs,opt} - T_{rs,min}} \frac{T - T_{rs,max}}{T_{rs,opt} - T_{rs,max}}$$
$$\frac{\partial q_{sec}}{\partial t} = \frac{\partial n_{sec}}{\partial t} x_{i,min} \tag{30}$$

with the multiplication factor $C_{RS} = 3.5 \cdot 10^8 \, \text{kg}^{-1}$ as well as the optimal temperature $T_{rs,opt} = 268 \, \text{K}$, the minimum temperature $T_{rs,min} = 265 \, \text{K}$ and the maximum temperature for rime splintering $T_{rs,max} = 270 \, \text{K}$. The rimed mass $q_{rime}$ represents the mass content of cloud droplets $q_c$ converted into graupel $q_g$ and hail $q_h$. For riming, the model considers the collision of supercooled droplets with ice crystals, snow, graupel and hail. The triangular profile is centered at $T_{rs,opt}$ and ejects up to 350 splinters per mg rimed ice mass. The fragments are initialized as SIP of the minimum ice mass $x_{i,min} = 10^{-12} \, \text{kg}$. Additional sources for SIP from raindrop freezing and shattering as well as collisional breakup will be presented in another study.

## 2.3 Depositional growth of ice particles

Deposition of water vapor onto a cloud particle in a supersaturated environment is the most fundamental growth process for all cloud particles. The SB scheme uses saturation adjustment for the water phase, hence whenever the environment is supersaturated wrt water ($S_w > 1$) at the beginning of a mircophysical time step, $S_w$ is relaxed to water saturation (i.e. $S_w \equiv 1$) and the change of water vapor $q_v$ is converted to cloud water $q_c$, considering latent heat release (using a Newton-Raphson scheme). However, supersaturation wrt ice (i.e. $S_i > 1$) is explicitly resolved. The mass growth of an ice particle by deposition of water vapor can be described by formulating the flux of vapor and heat between the particle and the environment considering mass and energy conservation, than integrating this fluxes over the particle surface and evaluating the effect of latent heat of sublimation on the particle surface temperature. Assuming that the temperature difference between the particle surface and the environment is small we can derive the general growth equation of a single ice particle with mass $x$ using the Clausius-Clapeyron relation (Pruppacher and Klett, 1998)

$$\left. \frac{\partial x}{\partial t} \right|_{dep} = \frac{4\pi C(x)(S_i - 1)F_v(x)}{\frac{T R_v}{D_v e_{si}} + \frac{L_s \left( \frac{L_s}{R_v T} - 1 \right)}{K_a T}} \tag{31}$$

with the saturation ratio wrt to ice $S_i = \frac{e}{e_{si}(T)}$, the ventilation coefficient $F_v$ and the capacitance $C$ accounting for the enhancement of the depositional growth by the flow field and the non-spherical shape, respectively. The left term in denominator represents the mass flux relation with the diffusion coefficient $D_v$ and water vapor saturation pressure wrt ice $e_{si}$; the right term describes the heat flux with the thermal conductivity of air $K_a$ and the latent heat of sublimation $L_s$. The capacitance can be related to the maximum diameter $C(x) = D(x)/c$ with $c_k$ depending on the class of the particle $k$.

Integrating (31) over the entire particle distribution for an ice mode $k$ we obtain

$$
g_{dep,k} = \left.\frac{\partial q_k}{\partial t}\right|_{dep} = \int_0^\infty \left.\frac{\partial x_k}{\partial t}\right|_{dep} \mathrm{d}x_k
$$

$$
= \frac{4\pi(S_i-1)c_k^{-1}}{\frac{TR_v}{D_v e_{si}} + \frac{L_s\left(\frac{L_s}{R_v T}-1\right)}{K_a T}} \int_0^\infty D(x_k)F_v(x_k)f(x_k)\,\mathrm{d}x_k \tag{32}
$$

$$
= \frac{4\pi(S_i-1)c_k^{-1}}{\frac{TR_v}{D_v e_{si}} + \frac{L_s\left(\frac{L_s}{R_v T}-1\right)}{K_a T}} D(\overline{x_k})\overline{F_v}
$$

with the mean particle mass $\overline{x_k}$ and averaged ventilation coefficient $\overline{F_v}$; for details, see calculations in the Appendix of SB. However, we must consider that all ice particle classes compete for water vapor. Thus we follow Morrison et al. (2005) semi-analytic approach to estimate the depositional growth rate with an exponential relaxation towards equilibrium. The change of mass content of ice particle class $k$ within a physical time step is

$$
\left.\frac{\Delta q_k}{\Delta t}\right|_{dep} = \frac{\delta_i X}{\tau_k}\left(1 - \exp\left(-\frac{\Delta t}{X}\right)\right) \tag{33}
$$

with $\delta_i = q_v - q_{si}$ expressing the supersaturation wrt ice and the microphysical time step $\Delta t$. $\tau_k$ is defined as the depositional time scale

$$
\tau_k^{-1} = \frac{g_{dep,k}}{\delta_i} \tag{34}
$$

$X$ is the conjoined relaxation time needed to describe the competition between the ice particle classes as a sum of the individual relaxation time scales

$$
X = \left[\sum_j^N \tau_j^{-1}\right]^{-1} \tag{35}
$$

where we consider graupel, hail, snow as well as all five ice modes. For a derivation and application of this method to resolve supersaturation see, e.g., Khvorostyanov (1995), Morrison et al. (2005), Köhler and Seifert (2015).

## 2.4    Sedimentation of cloud particles

For the sedimentation of ice particles, we use the mass-velocity relation (5) for ice crystals; these relations were used for all
new ice modes. For the use in the evolution equation of moments (6), an averaged sedimentation velocity must be calculated via a weighted integration, i.e.

$$
\bar{v}_{i,k} = \frac{1}{M_i^k}\int_0^\infty x^k f_i(x)v_i(x)\,\mathrm{d}x \tag{36}
$$

Using the analytical results for generalized gamma-distributions, this results into the expression

$$
\bar{v}_{i,k} = \alpha_i \frac{\Gamma\left(\frac{k+\nu_i+\beta_i+1}{\mu_i}\right)}{\Gamma\left(\frac{k+\nu_i+1}{\mu_i}\right)}\left[\frac{\Gamma\left(\frac{\nu_i+1}{\mu_i}\right)}{\Gamma\left(\frac{\nu_i+2}{\mu_i}\right)}\right]^{\beta_i} \bar{x}_i^{\beta_i} \tag{37}
$$

see also Section 3.7 in SB.

## 2.5 Collision processes of cloud particles

Collision is an important process for producing large cloud particles. For the formation of large raindrops, collision is essential. However, considering the formation of rain via the ice phase, again collisions between water and ice particles play an important role. In the presented scheme, the rates for collisions between liquid particles (cloud droplets and raindrops) remain unchanged.

However, the collision rates for ice particles must be reconsidered.

As formulated in the SB scheme, we have the following conceptual treatment for sorting new particles stemming from the collision of ice particles with others:

- Collision between a cloud ice particle and a cloud droplet leads to a larger ice particle or to a graupel particle, depending on the size of the ice particle (riming)

- Collision between two cloud ice particles leads to a snow particle (self-aggregation)

- Collision between a cloud ice particle and a snow particle leads to a snow particle (aggregation)

This concept must be adapted for ice particles from the five different ice modes. Especially, the collision of ice particles from different ice modes must be taken into account as a new process, extending the existing formulation of self-aggregation of ice particles. In the following the changes are documented.

### 2.5.1 Collision of liquid and solid particles


We adapt the formulation by SB of riming (ice particle collides with cloud droplet or raindrop) for all new ice modes. Analog to the implementation for a single ice class in SB, all ice modes can be collected by graupel, hail or snow and contribute to riming.

### 2.5.2 Aggregation of ice particles

There is no separate treatment for each ice mode regarding aggregation, all collisions between ice modes as well as self collection within a single ice mode contribute to the same snow class. The collision processes between two ice classes $i$ and $j$ leads to the formation of snow $s$ can be described as

$$\frac{\partial f_i(x)}{\partial t}\bigg|_{coll,ij} = -\int_0^\infty f_i(x)f_j(x')K_{ij}(x,x')\,\mathrm{d}x' \tag{38}$$

$$\frac{\partial f_j(x)}{\partial t}\bigg|_{coll,ij} = -\int_0^\infty f_i(x')f_j(x)K_{ij}(x,x')\,\mathrm{d}x' \tag{39}$$

$$\frac{\partial f_s(x)}{\partial t}\bigg|_{coll,ij} = \int_0^x f_i(x')f_j(x-x')K_{ij}(x',x-x')(x-x')\,\mathrm{d}x' \tag{40}$$

The collection kernel is defined as

$$K_{ij}(x_i,x_j) = A_{ij}E_{coll}|v_i(x_i) - v_j(x_j)| \tag{41}$$

where $A_{ij}$ is the cross section of the sweep out volume, $E_{coll}$ is the mean sticking efficiency and $v_{i,j}$ are the terminal velocities. Let $D_i$ and $D_j$ donate the maximum diameters of ice crystals and we obtain for $A_{ij}$

$$A_{ij}(x_i,x_j) = \frac{\pi}{4}(D_i(x_i) + D_j(x_j))^2 \tag{42}$$

where we simplified the expression by assuming spherical particles. The sticking efficiency is parameterised following Cotton et al. (1986)

$$E_{coll} = \min\left(10^{0.035\cdot(T-T_m)-0.7}, 0.2\right) \tag{43}$$

with a maximum efficiency of 0.2 for $T = 273\,\mathrm{K}$. The mass relation provided in equation (5) is not sufficient to characterise the terminal velocity of ice particles since complex shapes and atmospheric turbulence affect the flow field around the particle (Seifert, 2002). The velocity distribution function describes the probability of a ice particle of mass $x$ to have the terminal velocity $v'$

$$P(v'|x) = \frac{1}{\sigma_{vel}\sqrt{2\pi}}\exp\left[-\frac{1}{2}\left[\frac{v'-v(x)}{\sigma_{vel}}\right]^2\right] \tag{44}$$

with the variance of terminal velocity $\sigma_{vel}$.

The absolute velocity difference in the collision kernel would split the integral and impose incomplete gamma functions thus complicating the result. We use the Wisner approximation which assumes characteristic mean values for the terminal velocities that are constant and can hence be separated from the integrand. As proposed in Seifert (2002) integral cross-section is used as a weight

$$\overline{v_{ij,k}}^2 = \frac{1}{N_k}\int_{x_i=0}^{\infty}\int_{x_j=0}^{\infty} D_i^2(x_i)D_j^2(x_j)f_i(x_i)f_j(x_j)x_i^k$$
$$\int_{v_j=-\infty}^{\infty}\int_{v_i=-\infty}^{\infty}\left(v_i'(x_i) - v_j'(x_j)\right)^2 P(v_i'|x_i)P(v_j'|x_j)\,\mathrm{d}x_i\mathrm{d}x_j\mathrm{d}v_i'\mathrm{d}v_j' \tag{45}$$
$$= \overline{v_i^2}\vartheta_i^k + \overline{v_iv_j}\vartheta_{ij}^k + \overline{v_j}^2\vartheta_j^0 + 2\sigma_{vel}^2$$

where we used the properties of the Gamma function and the scaling factor

$$N_k = \int_{x_i=0}^{\infty}\int_{x_j=0}^{\infty} D_i^2(x_i)D_j^2(x_j)f_i(x_i)f_j(x_j)x_i^k\,\mathrm{d}x_i\mathrm{d}x_j \tag{46}$$

Integrating equations (38) to (40) and again using the properties of the Gamma function we finally obtain an analytical solution for the collision rate

$$
\begin{aligned}
\left.\frac{\partial n_s}{\partial t}\right|_{coll,ij} &= -\left.\frac{\partial n_i}{\partial t}\right|_{coll,ij} = -\left.\frac{\partial n_j}{\partial t}\right|_{coll,ij} \\
&= \frac{\pi}{4} E_{coll} n_i n_j \left[ \overline{D_i}^2 \delta_i^0 + \overline{D_i D_j} \delta_{ji}^0 + \overline{D_j}^2 \delta_j^0 \right] / \overline{v_{ij,0}}
\end{aligned}
\tag{47}
$$

and the change of mass contents due to aggregation

$$
\left.\frac{\partial q_i}{\partial t}\right|_{coll,ij} = -\frac{\pi}{4} E_{coll} q_i n_j \left[ \overline{D_i}^2 \delta_i^1 + \overline{D_i D_j} \delta_{ji}^1 + \overline{D_j}^2 \delta_j^0 \right] / \overline{v_{ij,1}}
\tag{48}
$$

$$
\left.\frac{\partial q_j}{\partial t}\right|_{coll,ij} = -\frac{\pi}{4} E_{coll} q_j n_i \left[ \overline{D_i}^2 \delta_i^0 + \overline{D_i D_j} \delta_{ij}^1 + \overline{D_j}^2 \delta_j^1 \right] / \overline{v_{ji,1}}
\tag{49}
$$

$$
\left.\frac{\partial q_s}{\partial t}\right|_{coll,ij} = -\left.\frac{\partial q_i}{\partial t}\right|_{coll,ij} - \left.\frac{\partial q_j}{\partial t}\right|_{coll,ij}
\tag{50}
$$

See Appendix C for the notation of the coefficients. Note that with the inclusion of 4 additional cloud ice classes, the number of collision processes contributing to snow have increased. There are 4 additional calls to ice self collection and 10 new calls to collisions between ice modes. The equations of the collision rates introduced in this section are in general non-linear. Hence the increased number of collision processes might affect the amount of snow in the model. We will analyse this bias (compared to the standard SB scheme) in Section 4.2.

## 3 Methods

Idealised simulations of a deep convective cloud were conducted with the Icosahedral Nonhydrostatic Weather and Climate Model (ICON) version 2.6.

### 3.1 Model setup

The setup of the ICON model largely follows the setup described in Heinze et al. (2017). ICON is run in limited area mode with constant boundary conditions derived from the initial conditions. The initial conditions are provided by the Weisman-Klemp setup described in Section 3.2. The domain extends over $2°$ in longitude and latitude without topography.

The model grid is an unstructured triangular grid, R02B11, with 2 initial divisions and 11 bisecting iterations (Wan et al., 2013). The effective horizontal resolution of the grid is $\sim 1.23\,\text{km}$. The vertical grid contains 128 levels. The physical time 505 step is $2\,\text{s}$. The simulation time for each experiment is $240\,\text{min}$. Output is written every $10\,\text{min}$ and interpolated on a regular longtitude-latitude grid with a spacing of $0.005° \times 0.005°$.

The model uses parametrisations for subgrid scale orographic drag (Lott and Miller, 1997) and non-orographic gravity wave drag (Orr et al., 2010). The turbulence and vertical diffusion scheme is (Raschendorfer, 2001). The setup does not use a surface scheme. Radiation transfer physics are disabled. Convection is treated explicitly, i.e. all convection parameterisations (shallow 510 and deep) are switched off.

### 3.2 Weisman-Klemp setup

The Weisman-Klemp setup (Weisman and Klemp, 1982, WK82 hereafter) is a suitable test environment to evaluate the ice mode scheme microphysics as all microphysic classes and associated model routines are active and both mixed-phase as well as pure ice clouds, the latter mostly located in the anvil, are observed. The WK82 setup is frequently used to achieve confidence 515 in newly developed models (Zängl et al., 2015). It has also been utilized in many studies to investigate convection or cloud microphysics [see, e.g., Bluestein and Weisman (2000); Takemi (2006); Miglietta and Rotunno (2014); Rousseau-Rizzi et al. (2017); Miltenberger et al. (2020); Barrett and Hoose (2023)]. The WK82 setup is available in the ICON version 2.6 as one of the standard test cases.

The vertical profiles of temperature and relative humidity are chosen following WK82 to represent a typical sounding of a deep 520 convection event with a maximum water vapor mixing ratio near the surface of $0.014\,\text{kg}\,\text{kg}^{-1}$ and a surface temperature of $300\,\text{K}$. The horizontal wind $u_z$ varies with altitude from 0 to $U_s = 5\,\text{m}\,\text{s}^{-1}$ Convection was initiated with a symmetric thermal perturbation of $1400\,\text{m}$ vertical extent and $10000\,\text{m}$ diameter with a temperature amplitude of $2\,\text{K}$ at the center, that decays to $0\,\text{K}$ at the edge of the bubble. Temperature, wind and moisture profiles of the setup can be found in WK82.

### 3.3 Experiments

Experiments with the ice modes scheme in several configurations were performed to evaluate the impact of different heterogeneous nucleation schemes on the distribution of the ice modes and the liquid origin vs in-situ formation pathway. Table 1 lists

| Name | Experiment |
|------|-----------|
| HA | HA15 het. ice nucleation |
| UL | UL18 het. ice nucleation |
| PH | PH08 het. ice nucleation |
| PH3 | PH08 het. ice nucleation with soot and bio mode |
| REF | reference simulation with original SB scheme |

**Table 1.** List of sensitivity experiments for idealised simulations

all experiments. HA, UL and PH label experiments using the heterogeneous ice nucleation schemes of Hande et al. (2015), Ullrich et al. (2017) and Phillips et al. (2008), respectively. The choice of parameters for the dust aerosol profiles are explained in Section 2.2.3. PH3 refers to a simulation with the Phillips et al. (2008) parametrisation including the soot and biological aerosol modes. The HA experiment denotes the default setup for the ice modes scheme as the Hande et al. (2015) is the most widely used heterogeneous ice nucleation scheme for ICON simulations with the SB microphysics scheme.

The reference simulation (REF) uses the standard SB scheme with only a single class for cloud ice. However, the same microphysical parametrisations and assumptions as the ice mode microphysics scheme are employed. In particular, REF uses the same setup for the heterogeneous ice nucleation as the HA experiment. Thus we can assess the ramifications of using multiple ice classes instead of a single one for the overall cloud evolution.

### 3.4 Ice cloud origin

We investigate the origin of cloud ice by the introduction of the ice mode mass fractions $f_x$ for ice mode $x$

$$f_x = \frac{q_x}{q_{tot}} \tag{51}$$

Thus $f_x$ describes the relative contribution of ice mode x to the total cloud ice at each grid point. Insignificant trace amounts of cloud ice can distort the statistics of ice mode mass fractions. Hence, we only calculate for mass fractions for grid points where the (total) cloud ice mass content exceeds $0.1\,\mathrm{mg\,kg^{-1}}$. This threshold value was used by previous studies that investigate ice cloud origins to identify regions were ice clouds of significant thickness are present (Wernli et al., 2016).

$f_{liq}$ is the liquid origin mass fraction describing the ratio of cloud ice stemming from liquid origin formation processes

$$f_{liq} = \frac{q_{frz} + q_{imm} + q_{sec}}{q_{frz} + q_{imm} + q_{hom} + q_{dep} + q_{sec}} \tag{52}$$

where HOM and DEP contribute to in-situ formed cirrus and FRZ and IMM to liquid origin cirrus.

### 4 Results

In this section we present the results of the idealised simulations of deep convection. In Section 4.1 we discuss the evolution and spatial distribution of hydrometeor classes and the ice modes in particular for the default ice modes experiment (see Section

3.3). In Section 4.2 we compare the HA and REF simulations to validate the new ice modes scheme against the established SB scheme. In Sections 4.3 we investigate the impact of different heterogeneous ice nucleation schemes on the ice modes by comparing simulations HA, UL, PH and PH3. In Section 4.4 we address the research question of liquid vs in-situ ice formation in the convective cloud.

## 4.1 Ice modes simulation

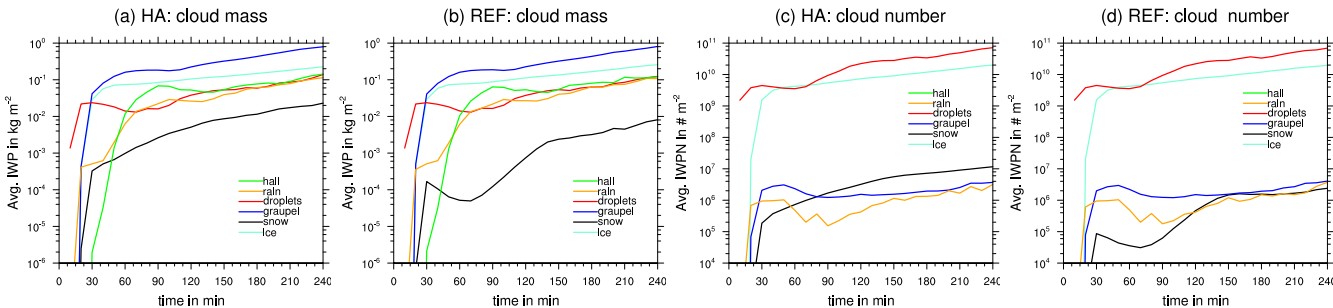

**Figure 3.** Temporal evolution of the horizontally averaged ice water path for each hydrometeor class for (a) ice modes (HA) and (b) reference SB simulation (REF). Temporal evolution of average integrated number concentration for (c) HA and (d) REF. ICE is the sum of all ice modes for HA.

Figure 3 show the temporal evolution of the different hydrometeor classes as the domain averaged (Ice) Water Path (I)WP (a,b) and vertically integrated number concentrations (I)WPN (c,d) beneath the cloud system. Ice is displayed as the sum of all ice modes (TOT). Note that the output interval of the model is $10\,\mathrm{min}$ and we only display domain averaged Water Path values above $10^{-6}\,\mathrm{kg\,m^{-2}}$. Hence the plots only show the time where hydrometeor classes appear in a 'significant' amount.

First we will focus on the experiment with the default setup for the ice modes scheme (HA) with Figure 3 Panels (a) and (c). Panels (b) and (d) will be discussed in Section 4.2. In Panel (a) we note that cloud droplets condense within the first $10\,\mathrm{min}$ of the simulation followed by the initiation of the warm rain process. Ice and graupel appear at the same time around $18\,\mathrm{min}$ and become quickly, within $5\,\mathrm{min}$, the two major contributors to the Water Path. Snow becomes relevant with a short delay of up to $5\,\mathrm{min}$ after the occurrence of graupel. Strong riming is the last microphysical process to become active with hail first occurring at $30\,\mathrm{min}$. At $60\,\mathrm{min}$ simulation time the cell matures. Hence, all hydrometeor classes are present and their avg. water path increases only slightly after that. Note that the cell shows no signs of dissipation within the simulation time. The ratio of the avg. water path between the classes also remains constant. Graupel followed by cloud ice are the dominate classes with regards to IWP. While hail, rain and cloud droplets show mostly the same avg. IWP values. Snow is distinctly the weakest hydrometeor class with regards to avg. IWP. The mass content of cloud particles in general is primarily governed by the availability of water vapour and thus thermodynamics. The number concentrations of the classes on the other hand are more varied since they directly depend on the formation pathways. Panel (c) in Figure 3 shows the avg. integrated number concentrations (IWPN) of all hydrometeor classes. We observe a large gap in avg. IWPN between cloud droplets and ice to the other classes with an

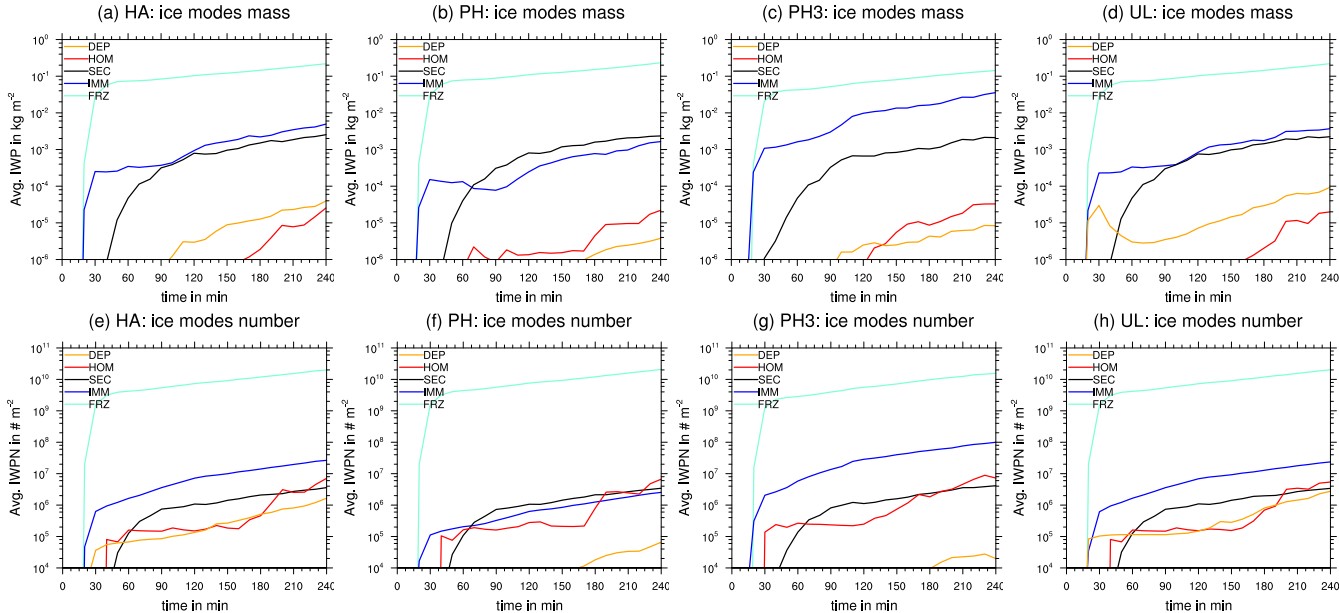

**Figure 4.** Temporal evolution of the horizontally averaged (ice) water path for each ice for heterogeneous ice nucleation scheme (a) HA, (b) PH, (c) PH3 and (d) UL. Average integrated number concentration for each ice mode for e) HA, (f) PH, (g) PH3 and (h) UL.

increase of 3.2 and 4 orders of magnitude for simulations time after $130\,\mathrm{min}$, respectively. Cloud droplet number concentrations are tied to the CCN (cloud condensation nuclei) activation scheme of Hande et al. (2016), which depends mainly on upward vertical velocity and thus produces a large droplet number concentration in a strong convective case like the WK82 setup. However, the scheme is still limited by the number of available CCN. We will discuss the connection between high cloud ice and cloud droplet number concentrations below.

Figure 4 provides insight into the development of each ice mode as domain averaged IWP and IWPN. We will focus on the results of the default setup of the ice mode schemes (HA) first (Panels (a) and (e)). At the beginning, ice consists of frozen cloud droplets where both immersion and homogeneous freezing, represented by the IMM and FRZ mode with avg. IWP values of $2 \cdot 10^{-4}$ and $5 \cdot 10^{-1}\,\mathrm{kg\,m^{-2}}$, respectively. FRZ is the most dominant ice mode by several orders of magnitude, especially with regards to avg. IWPN. Secondary ice from rime splintering (SEC) occurs with a $20\,\mathrm{min}$ offset to the first occurrence of graupel and becomes almost as important as immersion freezing in regards to avg. IWP with values up to $2 \cdot 10^{-3}\,\mathrm{kg\,m^{-2}}$ at simulation end. This is consistent with the findings of Miltenberger et al. (2020), who also performed simulations with the ice mode schemes for the same test case, but utilized lagrangian trajectory analysis to investigate rime splintering and spread of secondary ice through the cloud system. The first occurrence of secondary ice of significant number concentrations ($n_{sec} > 0.1 l^{-1}$) was observed after $30\,\mathrm{min}$ even for simulations of higher wind shear. They found that riming with graupel that sedimented from higher levels into the Hallett-Mossop temperature zone ($265 < T < 270\,\mathrm{K}$) to be the dominant source of secondary ice.

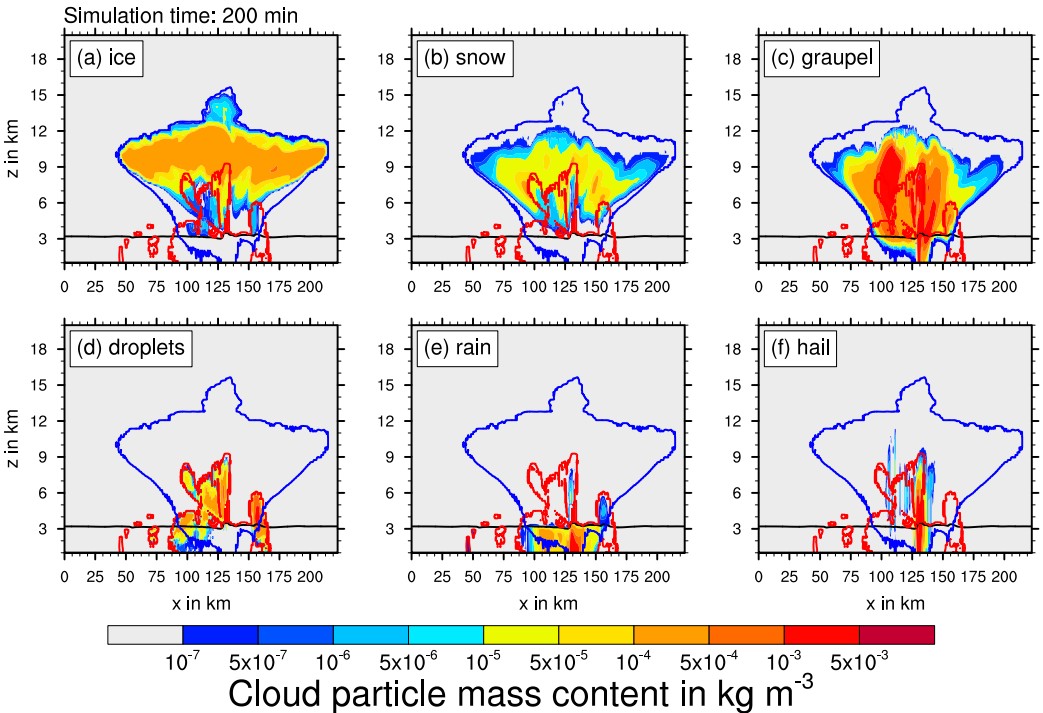

**Figure 5.** Vertical slice through domain center at $200\,\mathrm{min}$ simulation time. Mass content of (a) total ice, (b) snow, (c) graupel, (d) cloud droplets, (e) raindrops and (f) hail. The black line indicates the melting temperature level. The red and blue contour show the critical LWC and IWC value of $0.1\,\mathrm{mg\,m^{-3}}$, respectively.

Ice from homogeneous (HOM) and deposition nucleation (DEP) occurs first from nucleation events in the overshoot starting at $30\,\mathrm{min}$ and $40\,\mathrm{min}$ for DEP and HOM, respectively. Both types of nucleation events occur multiple times during the simulation and the avg. IWP of both modes increases over time. Initially, DEP is overall stronger in terms of avg. IWP although with significant IWP ($\geq 10^{-6}\,\mathrm{kg\,m^{-2}}$) occurring at $95\,\mathrm{min}$. However, at the end of the simulation HOM shows almost the same avg. IWP and IWPN values as DEP with $2\cdot 10^{-5}\,\mathrm{kg\,m^{-2}}$ and $2\cdot 10^{6}\,\mathrm{m^{-2}}$, respectively.

In summary, FRZ dominates with regards to avg. IWP with up to 2.5 orders of magnitude and with regards to avg. IWPN with up to 4 orders of magnitude. These results are strongly tied to CCN activation scheme. Droplets freeze, preferably, heterogeneously (here by immersion freezing) and then homogeneously once the active INPs for immersion freezing are depleted. Thus, if more CCN than INP are available, FRZ mode becomes more dominant than IMM as more (activated) droplets will freeze homogeneously once the INP are depleted. The integrated total ice number concentration (as the sum of all ice modes) is overall dictated by FRZ in this case and thus tied to the high number concentrations of droplets that the CCN scheme activates.

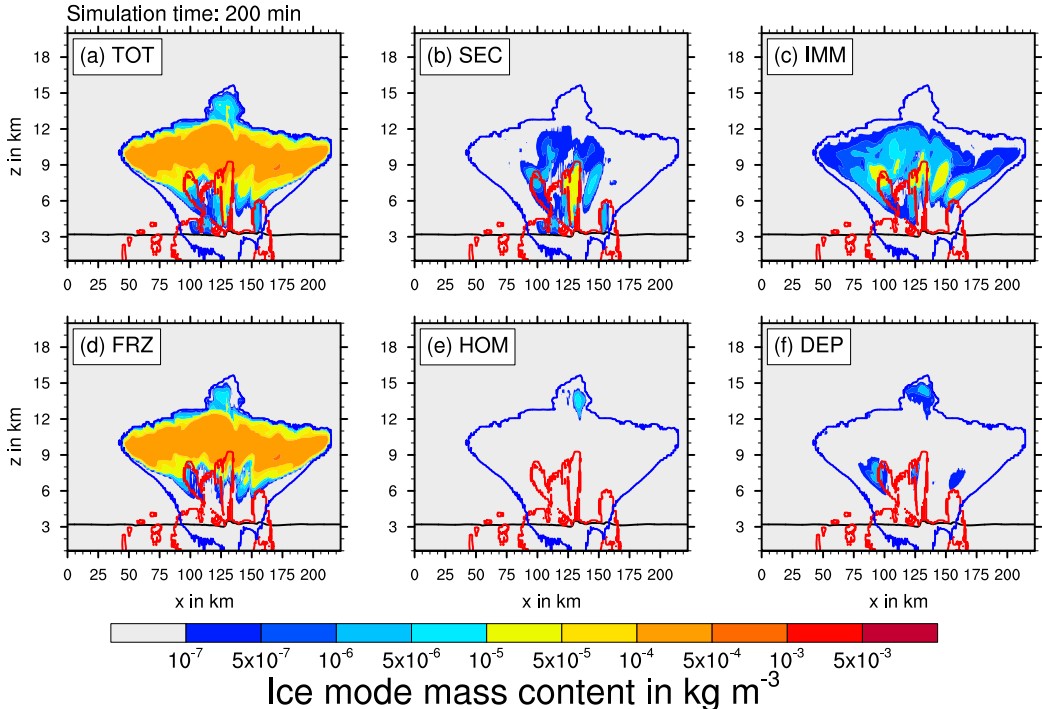

**Figure 6.** Vertical slice through domain center at $200\,\mathrm{min}$ simulation time. Mass content of (a) sum of all ice modes, (b) secondary ice, (c) heterogeneously frozen droplets, (d) homogeneously frozen droplets, (e) homogeneous nucleation and (f) deposition nucleation. The black line indicates the melting temperature level. The red and blue contours show the critical LWC and IWC value of $0.1\,\mathrm{mg\,m^{-3}}$, respectively.

We ill now investigate the vertical distribution of cloud particles. Figure 5 shows a vertical slice of the cloud through the domain center after $200\,\mathrm{min}$ simulation time and the mass content of all hydrometeor classes, where ice is represented as the sum of all ice modes. The red and blue contours show the critical LWC and IWC value of $0.1\,\mathrm{mg\,m^{-3}}$, respectively. The matured convective cell has a vertical extent of $15\,\mathrm{km}$ and developed an anvil between 9 and $12\,\mathrm{km}$. The cell produces strong precipitation beneath the cell as can be seen from Panel (e) for rain. Most of the cloud is glaciated with rimed particles, mainly graupel, being the most prevalent in terms of mass with mass contents of up to $10^{-3}\,\mathrm{kg\,m^{-3}}$. Hail is mostly found in the narrow updraft core at $130\,\mathrm{km}$, where riming rates and resident times are highest. Ice is present in most parts of the cloud above the melting temperature and forms an anvil of (pure) ice with an overshooting top above $12\,\mathrm{km}$. Snow is a product of aggregation and thus depends on the mass content and number concentrations of ice. Cloud regions with high cloud ice number concentrations and mass content have high aggregation rates. However, snow has a lower mean size than cloud ice and thus higher sedimentation rates. Thus, snow is mostly located below the maximum cloud ice mass content.

Figure 6 shows the same vertical slice as in Figure 5 but for the mass content of each ice mode after $200\,\mathrm{min}$ simulation time. The lower part of the cloud from 3 to $9\,\mathrm{km}$ around the updraft core is dominated by secondary ice from rime splintering

(SEC) with mass contents of up to $10^{-5}\,\mathrm{kg\,m^{-3}}$. This coincides with the region where riming is active, which is evident from high mass content of graupel and hail in these region as we observed in Figure 5. The prevalence of secondary ice from rime splintering in this region was also observed by Miltenberger et al. (2020).

Ice between 6 and $10\,\mathrm{km}$ is partially stemming from heterogeneously frozen droplets (IMM), while in the upper part of the cloud homogeneously frozen droplets (FRZ) are prevalent. The latter shows the highest mass contents for cloud ice with $10^{-4}\,\mathrm{kg\,m^{-3}}$ within a wide region extending from altitudes 8 to $12\,\mathrm{km}$ and the length of the convective cloud ($150\,\mathrm{km}$). When cloud droplets are transported upwards into colder regions they freeze first through immersion freezing due to the lower temperature threshold. Once all the INPs are activated, there is still a significant number of cloud droplets left or provided by the constant supply of humidity (and then saturation adjustment) from convection. At this point homogeneous freezing begins to dominate over immersion freezing. Also, the ice mode schemes only shows the ice formation pathway of the ice observed at this time, it does not track sources and sinks individually. That means that ice from IMM could have been converted to snow by aggregation or lost in collisions with graupel and is for that reason not as prevalent as FRZ.

The part of the cloud consisting of ice from FRZ stays structurally mostly the same during the simulation run time, only widening as the anvil expands. However, the cloud parts where IMM and SEC are present fluctuate with the evolution of the cell. SEC follows the liquid core where riming is occurring. IMM is at the beginning located above the liquid core between the sections were SEC and FRZ are prevalent and later mostly present in the flanks.

Ice from homogeneous nucleation (HOM) is limited to the overshoot, deposition nucleation (DEP) also occurs in small amounts outside the liquid core in the glaciated parts of the cloud between 6 and $9\,\mathrm{km}$.

Overall we notice a mostly clear spatial separation of the ice modes with some overlapping areas where ice of different origin is found. Ice in the upper part of the cloud above $9\,\mathrm{km}$ is for the majority of liquid origin, hence consisting of frozen droplets, and stemming from rime splintering below that. This general spatial distribution of the ice modes is also true for the other sensitivity experiments.

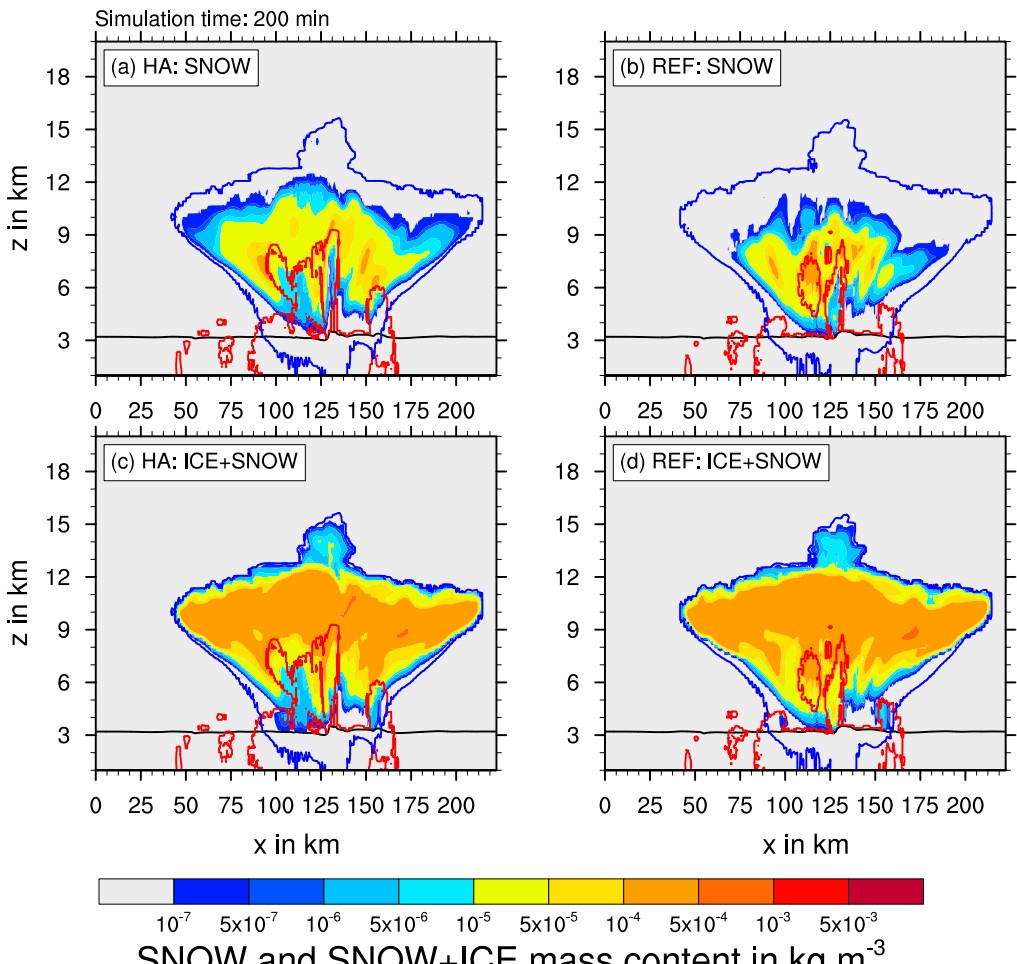

**Figure 7.** Vertical slice through domain center at $200\,\mathrm{min}$ simulation time. Mass content of snow for (a) ice modes scheme (HA) and (b) standard SB (REF) as well as the sum of all cloud ice and snow for (c) HA and (d) REF. The black line indicates the melting temperature level. The red and blue contour show the critical LWC and IWC value of $0.1\,\mathrm{mg\,m}^{-3}$, respectively.

## 4.2 Comparison - reference simulation

In this section we compare the reference simulation REF performed with the (unmodified) SB scheme to the ice modes simulation (HA). As elaborated in Section 3.3 both experiments use the same setup for heterogeneous nucleation (HA17). Thus the only difference between them is the split of the cloud ice class into the five ice modes.

The overall cloud evolution of the reference simulation (REF) is very close to the ice modes scheme (HA). As we can see in Figure 3 (b) the evolution of microphysic classes directly related to the liquid phase, that is cloud droplets, rain, graupel and hail, show the same temporal evolution. Even the onset times for cloud ice, where the governing parametrisations of both

schemes differ, are mostly the same. We observe, however, a large difference in the average Water Path of snow which shows an increase of an order of magnitude for the simulation with the ice modes scheme (HA). Although the difference decreases with time to a factor of $\sim 2.5$.

This also evident in the vertical slice of the convective cloud. Figure 7 shows the vertical slice of the convective cloud at $200\,\mathrm{min}$ through the domain center for snow with (a) the ice modes simulation and (b) the reference simulation. There is significantly more snow present in the ice mode simulation than in the reference simulation. While the maximum snow mass content is still at $10^4\,\mathrm{kg}\,\mathrm{m}^{-3}$, snow is distributed over a wider area of the cloud especially towards the flanks and higher altitudes (up to $11\,\mathrm{km}$). However, comparing the sum of ice and snow for both simulations in panels (c) and (d) shows that there is in general no increase of combined ice particle mass content. Rather we notice there is a tendency for the ice mode scheme (represented by the HA experiment) to shift mass from the ice to the snow class(es), hence the aggregations rate have to be higher. This is linked to the conceptional differences between both schemes: the SB standard scheme only produces snow by self collection of a single cloud ice class, where in the ice mode schemes, there are five independent cloud ice classes. They not only aggregate snow by self collection, but also by collisions with each other. While the physics of self collection and ice mode - ice mode collision are the same, the increased number of collisions processes leads to higher aggregation rates (see also Section 2.5.2.

While the shift of ice to snow should affect overall sedimentation rates of ice particles, there seems to be little impact on the cloud evolution and dynamics of the convective cell since the IWC and LWC outlines as well as the general evolution of the cloud observed in Figure 3 remains the same. It is also to note that pure ice and especially cirrus clouds will be mostly unaffected by that effect since collision efficiencies are small at low temperatures (see Section 2.5.2). Generally ice number concentrations are also insufficient to produce significant amounts of snow for pure ice clouds.

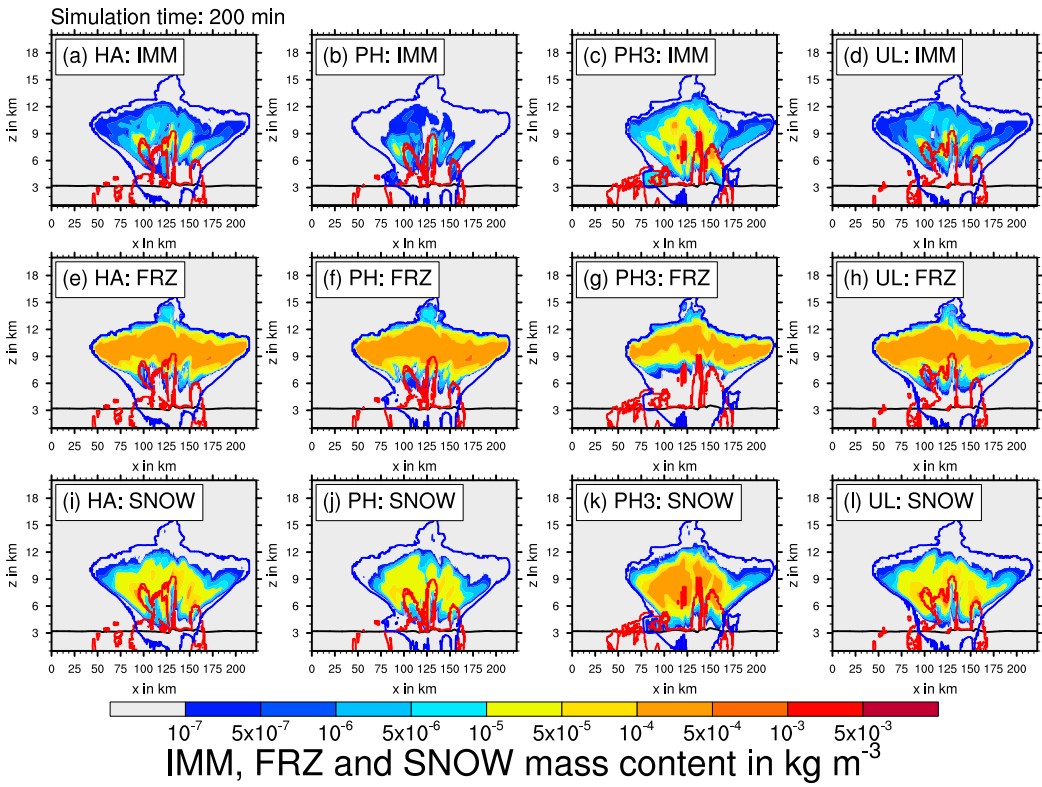

**Figure 8.** Vertical slice through domain center at $200\,\mathrm{min}$ simulation time comparing mass content of IMM (a-d), FRZ (f-h) and snow (i-l) for the experiments HA (a,e,i), PH (b,f,j), PH3 (c,g,k) and UL (d,h,l). The black line indicates the melting temperature level. The red and blue contour show the critical LWC and IWC value of $0.1\,\mathrm{mg\,m^{-3}}$, respectively.

### 4.3 Comparison - heterogeneous ice nucleation

Change of assumptions regarding distribution of INPs or a switch to a different INP activation scheme entirely can have a large impact on the evolution of the ice phase. We compare the experiments HA, PH, PH3 and UL here. For the general evolution of average Cloud Water Path and Ice Water Path, we do not observe major differences (not shown here). Hence, we will discuss the general evolution of the convective cloud only briefly here. Since the same CCN activation scheme is used for all experiments we do not observe significant change in the evolution of the liquid phase. The Rain Water Path is slightly lower for PH after the initial warm-rain formation, but catches up later. Hail formation starts sooner for PH but also reaches the same average value as for the other experiments. Further we observe a shift towards more snow and slightly less (total) cloud ice for the PH simulation. Thus the overall evolution of the convective clouds is similar for all heterogeneous ice nucleation schemes tested here.

We will now investigate the spatial and temporal distribution of the ice modes more closely for HA, PH, PH3 and UL. If we study the evolution of the ice modes average IWP in Figure 4 panels (a) to (d) we observe that FRZ and SEC stay mostly the same for all runs at avg. IWP of $10^{-3}$ and $10^{-1}\,\mathrm{kg\,m^{-2}}$, respectively. SEC stems from rime splintering and thus depends on the evolution of the graupel and hail class, which again is mostly sensitive to CCN activation and the resulting number of super-cooled droplets. We observed in Section 4.1 that SEC occupies a distinct part of the lower cloud around the updraft core. A change in heterogeneous nucleation scheme seems not to significantly affect the dominance of ice from rime splintering in this region or change the avg. IWP from secondary ice.

FRZ is also strongly sensitive to the CCN activation scheme which remains unchanged between the simulations. A weaker IMM mode should affect FRZ since droplets that do not freeze heterogeneously freeze homogeneously instead unless they are removed by evaporation or collision. Hence, both modes are in direct competition for unfrozen cloud droplets. However, FRZ is overall so dominant in this case study, that this effect is not noticeable with regards to avg. mass content and number concentrations.

Compared to HA, IMM is weaker in the simulation with the PH scheme both for avg. IWP and IWPN (Figure 4 (b,f)) with values of $5\cdot10^{-4}\,\mathrm{kg\,m^{-2}}$ and $10^{6}\,\mathrm{m^{-2}}$, respectively. However, for UL (panel (d,f)) avg. IWP and IWPN of IMM stays the same as for HA. The latter is not surprising since the assumptions about immersion freezing and the number concentrations of INPs are almost the same. In Section 2.2.3 we chose the dust concentration parameters for the PH parametrisation such that the maximum number of INPs activated for immersion freezing were the same as for the UL and HA schemes. The important difference of the PH to the HA and UL schemes is that immersion freezing occurs at higher temperatures (close to 263 instead of $258\,\mathrm{K}$). Thus, in lower and warmer parts of the cloud, where hydrometeor classes other than cloud ice are abundant, e.g. graupel. Collisions with those classes are an efficient sink for the IMM mode. Hence, the average IWP and IWPN of IMM are smaller for PH compared to HA and UL.

For PH3 (Figure 4 (c,h)) we observe higher avg. IWP of IMM than for all other simulations with values of up to $5\cdot10^{-2}\,\mathrm{kg\,m^{-2}}$. Even though we stated in Section 2.2.3 that the maximum amount of active INP is lower with this scheme than for the others, the inclusion of soot and biological aerosols triggers immersion freezing at higher temperatures even close to the melting

temperature level $T_m = 273\,\mathrm{K}$. Where for the PH scheme with the dust only mode, the shift to higher activation temperatures lowered the IMM content, for PH3, with the inclusion of soot and biological aerosol mode, IMM content is increased. Likely this even 'earlier' activation of INPs changes the cloud evolution such that the IMM mode becomes more important. Indeed, we will note that the cloud shape is different for PH3 when we later present a vertical slice of the cloud.

DEP is in generally weak in this convective case since deposition nucleation events usually only trigger outside mixed-phase clouds (see Section 2.2.3). In Figure 4 we observe different triggering times and strengths of deposition nucleation events. However, these differences are mostly caused by non-linear realisations of the model dynamics and not sensitive to the choice of nucleation scheme.

   While DEP is not continuously persistent in the overshoot, there are smaller nucleation events being triggered in the anvil
and the flanks of the convective cloud. Also note that output is written only every $10\,\mathrm{min}$, hence not all nucleation events are sampled in the output data and ice from DEP and HOM could be removed by aggregation or evaporation before they are sampled. When using the PH or PH3 scheme DEP is almost not present at all. This is consistent with our description of the scheme in Section 4.1, where even weak nucleation events are only triggered at high $S_i$ compared to UL and HA. Consequently, the HOM mode is strengthened for PH since homogeneous nucleation events are not suppressed by pre-existing ice from DEP.

Figure 8 shows vertical slices of the cloud at $200\,\mathrm{min}$ through the domain center. Plotted are the mass content of ice from immersion freezing (IMM) and homogeneously frozen droplets (FRZ) as well as snow for HA, PH, PH3 and UL (first, second, third and fourth column, respectively). As already discussed, FRZ dominates the upper part of the convective cloud with values of up to $10^{-4}\,\mathrm{kg\,m^{-3}}$ for all heterogeneous nucleation schemes. The structure of vertically layered ice modes described in Section 4.1 holds true for all simulations. As immersion freezing is mostly the same for HA and UL the distribution of ice
modes and snow shows only marginal differences between both experiments (first and fourth column).

   For PH, however, we found that IMM is much weaker in terms of mass content (and number densities). That also limits its horizontal distribution, which is confined more towards the core of the cloud compared to HA and UL. This directly affects the snow class, which shows lower number densities and less spreading throughout the cloud. This underlines again, that IMM is important for aggregation because it produces ice crystals large enough for efficient collection kernels (see Section 2.5.2).

Using PH3 affects not only the IMM mode but also the overall structure of the cloud. As discussed above, PH3 favors immersion freezing and its biological and soot INP modes show an onset for higher temperatures compared to other heterogeneous ice schemes. We observe the result of INP activation at higher temperatures with a shift of ice mass towards the lower model levels. This can also be observed in the shape of the convective cloud. For example, the PH3 cloud shows a lesser developed left flank. Additionally there is a pocket of IMM ice above the melting temperature line which is not present for HA and UL,
for which immersion freezing is not possible in this temperature region. While the general structure and layering of ice modes is the same, the shape and location of liquid water zones change (as can be seen from the red LWC outlines) which has a direct effect on the SEC mode that in general follows the liquid core (where riming occurs) closely. The PH3 simulation also sees an increase in snow underlining the importance of aggregation as a sink for the (in PH3 stronger) IMM mode.

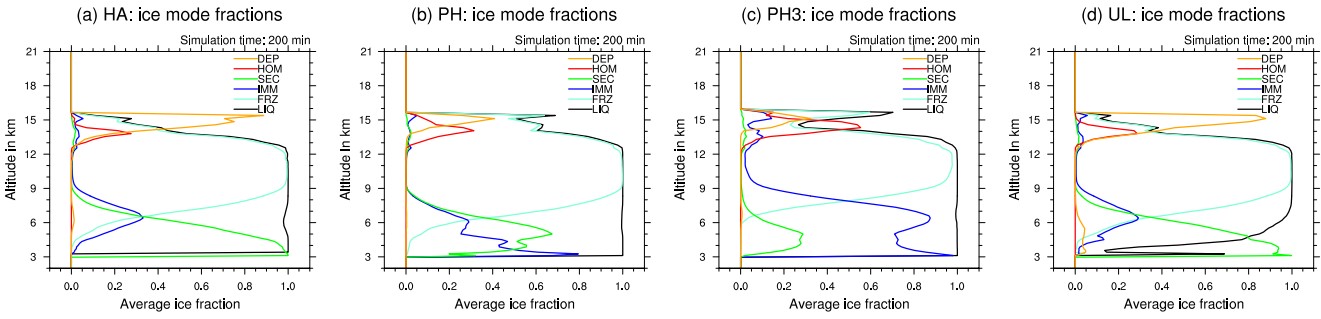

**Figure 9.** Average liquid origin and ice mode fractions for (a) HA, (b) PH, (c) PH3, (d) UL.

## 4.4 Liquid origin vs in-situ formation

Figure 9 shows the average mass fraction for all ice modes at $200\,\mathrm{min}$ simulations time. The mass fraction describes the ratio of ice from a particular mode $x$ to the total cloud ice $q_{tot}$ (see Section 3.4).

The majority of the fully glaciated parts of the cloud above the secondary ice region between 6 and $12\,\mathrm{km}$ is dominated by IMM (for HA, PH and UL) or FRZ (for PH3) classifying the parts of (pure) ice cloud as liquid origin (hence LIQ is close to 1). This includes the anvil of the deep convective cloud. The overshoot is located above $12\,\mathrm{km}$ and the liquid origin fraction

is determined by the amount of FRZ ice being mixed into this region and the strength of DEP nucleation events. PH and PH3 show a weak DEP mode with incursions from HOM resulting in liquid origin fractions between $0.3$ and $0.7$. For HA and UL the liquid origin fraction is below $0.4$. This makes the overshoot a region where ice stemming from different formation pathways mixes. The dynamic forcing transporting FRZ and IMM ice into the overshoot, the occurrence and strength of HOM and DEP nucleation events and the sedimentation of newly formed ice from the overshoot into the cloud below all affect and

change the liquid origin fraction. That the majority of the fully glaciated parts of the deep convection cloud are of liquid origin is in accordance with previous studies (Gasparini et al., 2018).

## 5 Discussion and Summary

We presented a new microphysics bulk model adapted from the Seifert and Beheng (2006) two-moment scheme by introducing multiple ice classes each with their unique particle formation mechanism: homogeneous freezing of solution and pure water droplets, immersion freezing, deposition nucleation and secondary ice from rime splintering. The microphysical processes governing these ice modes have been described with an emphasis on the particle formation mechanisms.

Idealised simulations of a convective cloud, using the Weisman-Klemp setup, were performed to validate the model with a comparison to a reference simulation using the standard SB scheme. The Weisman-Klemp test is a highly idealized description of the typical life cycle of a deep convective scenario. However, it contains the major features of the development of a convective storm, as e.g. the development of liquid clouds at lower levels, which are transformed into a mixed-phase clouds, and also the typical development of the anvil cloud. Although the scenario cannot be compared in a 1-1 way to measurements, it contains the main features of the convective life cycle. Thus, we can study the formation and evolution of ice clouds exemplary, and can derive insights into the general distribution of ice clouds and their formation pathways. Thus, the derived statistics can serve as a first, but still idealized, evaluation of ice clouds in convective storms, and nevertheless it gives first quantitative results about the distribution of formation pathways.

It was shown that the ice mode schemes reproduces the same cloud evolution for the dynamics and all particle classes except for snow, where we observed a shift of mass content from ice crystals to snow. This is linked to an increase of collision processes due to introduction of multiple ice classes.

We found that the scheme showed a reasonable distribution of the ice modes with liquid origin ice, formed of homogeneously and immersion frozen droplets, to constitute the majority of ice present in the matured cell. The occurrence of in-situ ice formed of homogeneous and deposition nucleation origin was, compared to other modes, low as it is expected for a convective cloud. Most in-situ ice modes were only present in the overshoot. But even there they mixed with liquid origin ice modes that were transported into the overshoot.

Simulations with four different heterogeneous nucleation parametrisations showed that the ice mode scheme provides the basis for a nuanced analysis to evaluate the impact of parametrisation choice not only on the total ice content and number concentrations but on the competition between ice particle formation pathways as well.

We investigated rime splintering as a secondary ice mechanism. In general we observed that it not significantly enhanced ice content in parts of the convective clouds were primary ice formation is active. Rather secondary ice was relevant in thermodynamic regions where primary ice formation is insufficient. Thus it helped to expand cloud ice especially towards higher temperature levels. However, we note that the importance and high number densities of secondary ice from RS in the lower parts of the cloud (below $7\,\mathrm{km}$) are concerning since the underlying parametrisation of Hallett-Mossop could not be confirmed by recent laboratory studies (Seidel et al., 2024). In a future work we will implement additional secondary ice processes, droplet shattering as well as collisional breakup, and compare them to rime splintering.

In the future we consider combining the ice modes schemes with tracking of (accumulated) microphysical process rates. As discussed in the introduction both viewpoints are complementary as it allows to combine information of region where ice

nucleation is active with the distribution of ice modes later. The ice mode scheme might 'miss' cloud ice that formed, e.g. by immersion freezing, but is quickly (within the output write steps) removed due to collision or evaporation. Another future model extension is implementing the ice mode schemes into an ICON version that includes the Aerosol and Reactive Trace gases (ART) model component, which allows explicit modelling of aerosol serving as ice nucleating particles.

In the next study we are going to investigate the origin of cloud ice in the cloud band and outflow region of Warm Conveyor Belts especially addressing the open research question of in-situ vs liquid origin cirrus. There we will also compare the classification of the ice modes scheme with other cirrus classification methods, e.g., Wernli et al. (2016).

*Code availability.* The NCL code for the data evaluation is available upon request.

*Data availability.* All data are available from the authors upon request.

**Appendix A: List of abbreviations**

| Abbreviation | Description |
| --- | --- |
| DEP | deposition nucleation ice mode |
| FRZ | homogeneous freezing of cloud droplets ice mode |
| HA15 | Hande et al. (2015) ice nucleation scheme |
| HOM | homogeneous freezing of solution droplets ice mode |
| IMM | immersion freezing ice mode |
| IWC | ice water content |
| IWP | ice water path |
| IWPN | vertically integrated ice number concentration |
| LWC | liquid water content |
| PH08 | Phillips et al. (2008) ice nucleation scheme |
| UL17 | Ullrich et al. (2017) ice nucleation scheme |
| SB | Seifert and Beheng (2006) two-moment scheme |
| SEC | secondary ice mode |
| SIP | secondary ice particle |
| TOT | sum of all ice modes |
| WK82 | Weisman-Klemp test setup |

**Table A1.** List of abbreviations

## Appendix B: Generalized Gamma distribution

The generalized gamma distribution is used to describe the mass or size distribution of a cloud particle class

$$f(x) = Ax^\nu \exp(-\lambda x^\mu) \tag{B1}$$

where we use the particle mass $x$ as the variable and the shape and scale parameters $\nu$ and $\mu$ are prescribed. $\lambda$ and $A$ are linked to the prognostic distribution moments, the zeroth and first moment, corresponding to the number concentration $M^0 = n$ and mass mixing ratio $M^1 = q$, respectively. To obtain the n-th mass weighed moment of the generalized gamma distribution $M^n$ we multiply equation (B1) with $x^n$ and integrate over the entire domain which yields

$$
\begin{aligned}
M^n &= \int_0^\infty f(x)x^n \, \mathrm{d}x = \int_0^\infty A x^{\nu+n} \exp(-\lambda x^\mu) \, \mathrm{d}x \\
&= \frac{A}{\mu} \int_0^\infty y^{\frac{\nu+n+1}{\mu}-1} \exp(-\lambda y) \, \mathrm{d}y \\
&= \frac{A}{\mu \lambda^{\frac{\nu+n+1}{\mu}}} \Gamma\left(\frac{\nu+n+1}{\mu}\right)
\end{aligned}
\tag{B2}
$$

where we used the substitution $y = x^\mu$ and the relation of the Gamma function $\int_0^\infty y^\zeta \exp(-\eta y) \, \mathrm{d}y = \Gamma(\zeta+1)\eta^{-(\zeta+1)}$ for $\zeta > -1$, which can be reduced to Euler's definition of the Gamma function with the use of a suitable substitution.

Prescribing constant values for the shape parameters $\nu$ and $\mu$ we can now solve equation (B2) for $A$ and $\lambda$

$$\lambda = \left(\frac{n}{q} \frac{\Gamma\left(\frac{\nu+2}{\mu}\right)}{\Gamma\left(\frac{\nu+1}{\mu}\right)}\right)^\mu \tag{B3}$$

$$A = \frac{\mu n}{\Gamma\left(\frac{\nu+1}{\mu}\right)} \lambda^{\frac{\nu+1}{\mu}} = \frac{\mu n}{\Gamma\left(\frac{\nu+1}{\mu}\right)} \left(\frac{n}{q} \frac{\Gamma\left(\frac{\nu+2}{\mu}\right)}{\Gamma\left(\frac{\nu+1}{\mu}\right)}\right)^{\nu+1} \tag{B4}$$

and so we can finally describe the distribution function as a function of particle mass and the two prognostic moments

$$f(x,q,n) = \left[\frac{x}{\overline{x}}\right]^\nu \frac{n}{\mu \overline{x} \Gamma\left(\frac{\nu+1}{\mu}\right)} \left[\frac{\Gamma\left(\frac{\nu+2}{\mu}\right)}{\Gamma\left(\frac{\nu+1}{\mu}\right)}\right]^{\nu+1} \exp\left(-\left[\frac{x}{\overline{x}} \frac{\Gamma\left(\frac{\nu+2}{\mu}\right)}{\Gamma\left(\frac{\nu+1}{\mu}\right)}\right]^\mu\right) \tag{B5}$$

with the mean particle mass $\overline{x} = \frac{q}{n}$.

## Appendix C: Notation collision integrals

These shorthand notations are used for the coefficients in the analytical results of the collision integrals in Section 2.5.2 and can be calculated beforehand since they only contain fixed parameters

$$\delta_i^k = \frac{\Gamma\left(\frac{2b_i+1+\nu_i+k}{\mu_i}\right)}{\Gamma\left(\frac{\nu_i+1}{\mu_i}\right)} \left(\frac{\Gamma\left(\frac{\nu_i+1}{\mu_i}\right)}{\Gamma\left(\frac{\nu_i+2}{\mu_i}\right)}\right)^{2b_i+k} \tag{C1}$$

$$\delta_{ji}^k = \frac{\Gamma\left(\frac{b_i+1+\nu_i+k}{\mu_i}\right)}{\Gamma\left(\frac{\nu_i+k}{\mu_i}\right)} \left(\frac{\Gamma\left(\frac{\nu_i+1}{\mu_i}\right)}{\Gamma\left(\frac{\nu_i+2}{\mu_i}\right)}\right)^{b_i+1} \frac{\Gamma\left(\frac{b_j+1+\nu_j}{\mu_j}\right)}{\Gamma\left(\frac{\nu_j+1}{\mu_j}\right)} \left(\frac{\Gamma\left(\frac{\nu_j+1}{\mu_j}\right)}{\Gamma\left(\frac{\nu_j+2}{\mu_j}\right)}\right)^{b_j} \tag{C2}$$

$$\vartheta_i^k = \frac{\Gamma\left(\frac{\nu_i+2b_i+1+2\beta_i+k}{\mu_i}\right)}{\Gamma\left(\frac{2b_1+1+\nu_i+k}{\mu_i}\right)} \left(\frac{\Gamma\left(\frac{\nu_i+1}{\mu_i}\right)}{\Gamma\left(\frac{\nu_i+2}{\mu_i}\right)}\right)^{2\beta_i} \tag{C3}$$

$$\vartheta_{ji}^k = \frac{\Gamma\left(\frac{\nu_j+2b_j+\beta_j+1}{\mu_j}\right)}{\Gamma\left(\frac{2b_j+1+\nu_j}{\mu_j}\right)} \left(\frac{\Gamma\left(\frac{\nu_j+1}{\mu_j}\right)}{\Gamma\left(\frac{\nu_j+2}{\mu_j}\right)}\right)^{\beta_j} \frac{\Gamma\left(\frac{\nu_i+2b_i+1+\beta_i+k}{\mu_i}\right)}{\Gamma\left(\frac{2b_i+1+\nu_i+k}{\mu_i}\right)} \left(\frac{\Gamma\left(\frac{\nu_i+1}{\mu_i}\right)}{\Gamma\left(\frac{\nu_i+2}{\mu_i}\right)}\right)^{\beta_i} \tag{C4}$$

*Author contributions.* TL and AS did the model development; TL and PS designed the study; TL performed the simulations and carried out the data analyses; TL and PS contributed to interpreting the results and writing the paper.

*Competing interests.* The contact author has declared that none of the authors has any competing interests.

*Acknowledgements.* The research leading to these results has been done within the subproject B7 of the Transregional Collaborative Research Center SFB/TRR 165 Waves to Weather funded by the German Research Foundation (DFG).

Parts of this research were conducted using the supercomputer MOGON 2 and/or advisory services offered by Johannes Gutenberg University Mainz (hpc.uni-mainz.de), which is a member of the AHRP (Alliance for High Performance Computing in Rhineland Palatinate, www.ahrp.info) and the Gauss Alliance e.V.

The authors gratefully acknowledge the computing time granted on the supercomputer MOGON 2 at Johannes Gutenberg University Mainz (hpc.uni-mainz.de).

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
