# Peer review of "Investigating ice formation pathways using a novel two-moment multi-class cloud microphysics scheme"

_EGUsphere, 2024_

## Author Comment (AC1)

This manuscript provides a detailed overview of the development of a new bulk microphysics scheme that includes several classes of ice hydrometeor, as opposed to the typical two classes in older schemes. It also includes the evaluation of a test case in an idealized convective cloud that clearly shows how differences between ice microphysical assumptions impact key properties/processes within a cloud. The further development of ice microphysical schemes is important to better understand how the ice phase influences key cloud properties. However, that message does not come across clearly enough in this work, with the sparse amount of . Similarly, I recommend that the authors undergo more rounds of revision and proofreading before this manuscript is ready for publication. I have included specific comments/questions/suggestions below to aid in the revision.

**Response:** We expanded the introduction with additional text discussing the benefits, applications and limitations of the new ice mode microphysics scheme. A comparison to other methods that can evaluate ice formation pathways, e.g., tracking of process rates and particle models, was also included.

**Introduction**

- Here is where I found that the discussion did not cover enough of the work to date to justify the need for a new multi-class scheme, with too little literature cited overall. A couple background topics to consider covering: What is it about two class schemes that make them unsuitable to simulate ice microphysics? What have others done with multi-class schemes and what have they found (consider reviewing the work on the P3 scheme by Morrison and Milbrandt, 2015 and others).

**Response:** We added a general introduction to bulk microphysic schemes and a list of most commonly used schemes in cloud research (including the P3 scheme). Also we added a comparison of the ice modes scheme to other methods that can be used to investigate ice formation pathways, e.g., super particle models and tracking of microphysical process rates.

**Model description**

- Lines 142-149: liquid-origin and in situ were already defined in the introduction so I don't think you need to redefine them here

**Response:** The second definition of the liquid origin and in-situ pathways was removed in the revised manuscript.

- Line 166: why does the saturation adjustment mean that the direct numerical integration of the homogeneous freezing coefficient?

**Response:** We rephrased the sentence. The parametrization for the nucleation rate for homogeneous freezing of cloud droplets is only valid at water saturation. However, the model is always at water saturation when cloud droplets are present due to saturation adjustment.

- Lines 243-245: the fact that the model does not include an explicit aerosol microphysics model for simulating CCN and INPs, this could be considered for future work

**Response:** An implementation of the ice modes scheme into ICON-ART, which features an explicit aerosol model, can be considered for future work. However, it is out of scope for this and the following studies. Introducing explicit aerosol microphysics requires the constrainment of aerosol sources and sinks. This introduces an additional large uncertainty for model initialization, aerosol transport and emissions.

Climatological aerosol profiles are well constrained and the standard option for cloud microphysics in operational and research model setups. We consider the current implementation of aerosol microphysics sufficient to investigate liquid origin vs in-situ ice formation.

- Figure 2: please readjust panel a such that xtick labels are not overlapping

**Response:** The issue got correct along side a general stylistic overhaul of figures.

- Section 2.3, consider renaming as the contents cover depositional growth of ice particles so the title may confuse readers

**Response:** Changed title of subsection into 'Depositional growth of ice particles'.

- Section 2.2.4: consider stating that the differences between these heterogeneous nucleation parameterizations are tested below for an idealized convective case because of the implications on the ice phase in mixed-phase clouds (as stated in the text)

**Response:** Added an extended introduction for the choices of heterogeneous nucleation schemes and a reference to their comparison in the result section (3.3).

**Idealised simulations**

- The authors should make it clearer in the text that they tested the new multi-class ice scheme with the three heterogene0ous nucleation options by HA, UL, and PH (with the additional PH3) as detailed in Section 2.

**Response:** We separated the list of experiments from the results section into the (new) method sections (now Section 3.) and added a new introduction to the results section (now Section 4.). This should make the setup of the experiments and the purpose of each of the result sections clearer.

- Please make clearer that the REF scheme refers to the SB scheme with only two ice modes

**Response:** We expanded the experiments subsection (now Section 3.3) with a clearer explanation of the differences and purposes of the REF and HA experiments.

- It is typical to explicitly state the model version, resolution, and domain (both space and time) up front in a description of experiments, or in the methods section before this. For example, there are several versions of the ICON model, so it should be made clear that the authors ran it in LES mode following Heinze et al. (2017). What other standard configurations are worth noting for your model version? This helps with reproducibility of your results.

**Response:** With the introduction of the methods section (now Section 3.) the description of the model setup was also expanded (Section 3.1). It contains now more information about the model grid, domain, initial conditions and all physical parametrizations.

- What is the Weisman-Klemp setup? Is it the prescribed temperature and relative humidity profile? Why this set up over others (if available)?

**Response:** The previous information about the Weisman-Klemp were now moved to a new section in methods (3.2). We also added references to other studies using the Weisman-Klemp setup for the investigation of (idealized) convection and general test of moist atmosphere physics.

- Why is the comparison made only between REF and HA at the start of Section 3.1?

**Response:** We added a new introduction to the results section (now Section 4.) which explains the purpose of each results section. The first step is to investigate the evolution and structure of the convective cloud for default setup of the ice mode microphysics (now Section 4.1). Then we compare the simulation of the new ice modes scheme (HA) with the reference simulation (REF) in Section (now) 4.2. The only difference in the setup between HA and REF is that HA uses 5 cloud ice classes and REF uses 1 class. The parametrization, especially for the heterogeneous ice nucleation, are the same. Thus we can use the comparison to validate the new microphysics scheme by comparison to the widely used SB scheme. Since REF uses only one ice class, it is reasonable to focus on the comparison of (total) cloud ice (as the sum of all ice modes for HA) and the other hydrometeor classes first (Figure 3.).

For the comparison of the heterogeneous ice nucleation schemes, all experiments were performed with the ice modes scheme. Thus we can focus on the differences in the ice modes.

- In section 3.1 it might be worth noting that you discuss the other panels in figure 3 later on

**Response:** We added clarification that only two panels are discussed at this point. We did the same for the first mention of Figure 4. in Section 4.1 (previously Section 3.1).

- Line 511, sentence starting with "However, it is heavier…". Does the "it" refer to ice, as in total ice mass? Maybe consider making this clearer.

**Response:** The referred lines were expanded and rewritten in the revised manuscript.

- Figures 5, 6, and 7 are generally great. Where did the 0.1 mg/m3 threshold come from?

**Response:** We added a reference to Wernli et al. (2016) in (new) Section 3.4 for the threshold value. In general, we consider clouds with a mass content of less than 0.1 mg/m3 as too thin to be significant. However, when included, they can distort ice mode mass fraction statistics as explained in Section 3.4.

- Lines 543-544 and Figure 7: "There is significantly more snow present in the ice mode simulation than in the reference simulation." That does not seem to be the case looking at figure 7. Can the authors explain what they mean more clearly?

**Response:** We fixed a mislabeling of the Figure 7 panels. In the revised figure, we note that snow in the HA experiment (Panel (a)) is spread over a wider area than in the REF experiment (Panel (b)). The (slight) increase in snow is also noticeable in the temporal evolution of average integrated snow mass content (compare black lines in Figure 3 (a) and (b)).

- Line 547, is this sentence referring to the HA scheme?

**Response:** We added a clarification. The as 'HA' labeled experiment represents the simulation with the ice mode schemes in this section. Both experiments, REF and HA, use the (Hande et al 17., 2015) heterogeneous ice nucleation scheme. The expanded introduction to the experiments in Sections 3.3, 4. and 4.2 should also clarify this.

- Line 588: please state which figure again, and I think the authors might mean panels c and g instead.

**Response:** We added a reference to the figure here.

**Discussion**

- Are there any limitations with this new scheme that weren't addressed by this study that could be considered in future work?

**Response:** Concerning the liquid origin vs in-situ research question, the ice mode schemes only evaluates from which microphysical formation process the ice crystals stem. It does not track the microphysical history of the ice crystals after their formation. However, the thermodynamic environment that the ice crystals are subject to and grow in (by deposition of vapor) might be as important as their formation process for macrophysical properties of (pure) ice clouds. We will investigate this in the next study.

**General**

- Several instances of using "an" where there should have been and "a" and vice versa

**Response:** We made a complete pass trough the manuscript to correct this type of mistake specifically.

- Several acronyms weren't defined before their use or the longname was used after the acronym was defined, check for consistency

**Response:** We made a pass trough the manuscript to ensure that acronyms are always defined before first used. They are also listed in the table of of abbreviations in Appendix A. We also corrected instances where the longnames were used in the same sections as the acronyms after their definition. However, for the readability of the manuscript we believe it to be helpful to reintroduce acronyms when they were not used for a several sections. Here we will confer with the editor for stylistic guidelines.

- Are there any observations of precipitation rates that the authors could compare to their new model setup as a test of their new scheme?

**Response:** We are not aware of a test case based on observations that allows to evaluate the performance of microphysics schemes. As detailed in Section 3.2 the Weisman-Klemp test is widely used as the standard test environment to evaluate microphysics schemes. The validation of the ice modes scheme in this study is done by comparison to the (unmodified) Seifert and Beheng (2006) double-moment scheme, which is widely used in cloud research (Section 4.2). In future studies we will employ the ice mode scheme for studies of real cases. This will include validation by comparison of the ice modes scheme to satellite observations.

**References**

Hande, L. B., Engler, C., Hoose, C., and Tegen, I.: Seasonal variability of Saharan desert dust and ice nucleating particles over Europe, Atmospheric Chemistry and Physics, 15, 4389–4397, https://doi.org/10.5194/acp-15-4389-2015, 2015.

Hande, L. B., C. Hoose, and C. Barthlott, 2017: Aerosol- and Droplet-Dependent Contact Freezing: Parameterization Development and Case Study. *J. Atmos. Sci.*, **74**, 2229–2245, https://doi.org/10.1175/JAS-D-16-0313.1.

Hulburt, H. M. and Katz, S.: Some problems in particle technology: A statistical mechanical formulation, Chemical engineering science, 19, 555–574, 1964

Kärcher, B. and Lohmann, U.: A parameterization of cirrus cloud formation: Homogeneous freezing of supercooled aerosols, Journal of Geophysical Research: Atmospheres, 107, AAC–4, https://doi.org/10.1029/2001JD000470, 2002

Kärcher, B., Hendricks, J., and Lohmann, U.: Physically based parameterization of cirrus cloud formation for use in global atmospheric models, Journal of Geophysical Research: Atmospheres, 111, https://doi.org/10.1029/2005JD006219, 2006.

Seifert, A. and Beheng, K. D.: A two-moment cloud microphysics parameterization for mixed-phase clouds. Part 1: Model description, Meteorology and atmospheric physics, 92, 45–66, https://doi.org/10.1007/s00703-005-0112-4, 2006.

Spichtinger, P. and K. M. Gierens (2009). "Modelling of cirrus clouds – Part 1a: Model description and validation". In: Atmospheric Chemistry and Physics.

Wernli, H., Boettcher, M., Joos, H., Miltenberger, A. K., and Spichtinger, P.: A trajectory-based classification of ERA-Interim ice clouds in the region of the North Atlantic storm track, Geophysical Research Letters, 43, 6657–6664, https://doi.org/10.1002/2016GL068922, 2016.

---

## Author Comment (AC2)

- **RC2**: ['Comment on egusphere-2024-2157'](), Anonymous Referee #2, 18 Oct 2024

This work introduces new ice classes into an existing cloud microphysics scheme to better elucidate the role of different ice formation mechanisms on the cloud evolution. Since the role of the ice phase on the cloud evolution is still fraught with uncertainty this work and its conclusions are of significance to the scientific community. It is also well written, however some issues must be addressed before publication, as described below.

**General comments.**

The introduction is almost completely devoid of any citations, despite being full of overarching generalizations. It makes me wonder whether the authors conducted a proper literature review before writing the paper. Please add citations and put the work within the context of the current literature.

It is important to recognize that this is a very idealized model, that although illustrative it might have limited application in the modeling of real situations. The size distribution of the ice crystals is a single function and can't be separated by origin as ice grows. It would be impossible to measure or even estimate the parameters of the distribution for the different ice classes. It is also not recommended to excessively add tracers to be advected by the host model since it may lead to numerical diffusion issues. For host models with lower resolution it would be difficult to relate the different classes to macro-scale variables like cloud fraction and total condensate. Finally, the INP classes used in the work are somehow artificially separated by design while in real cloud they tend to be active at the same time. These considerations must be made clear in the discussion section of the work.

**Response:** We addressed the concerns of the reviewer in the specific comments. Additional text was added in the 'Introduction' and 'Discussion and Summary' sections to explain the purpose and benefits of the new microphysics scheme and compare it to other methods for investigation of ice formation pathways.

**Specific comments**

Line 19. Clouds are uncertain in models not in the Earth, please rephrase.

**Response:** We rephrased the introductory sentences.

Lines 24, 29, 44, 45, 54, 55 and many other places. These statements are not obvious and need references backing them up.

**Response:** Additional references were added with the rewrite of the introduction.

Line 52. Cloud types.

**Response:** Fixed misspelling.

Line 59. Remove "will".

**Response:** Rephrased sentence.

Line 70. Please rewrite this sentence in clearer terms.

**Response:** An introduction to bulk microphysics schemes in general was added to the introduction.

Line 74. What is f? Should it be normalized to the total mass instead?

**Response:** Integrating over the size distribution from 0 to infinity we obtain the total number density of particles in the reference volume. Thus, the size distribution is normalized to total number density (or total number concentration if multiplied by air density).

Line 76. Define "the particle mass distribution".

**Response:** A size distribution describes the number density of particles as a function of the phase space (Hulburt and Katz, 1964). Integrating over the size distribution we obtain the number density (or number concentration if multiplied by the air density) in an interval of the phase space, e.g., in the interval from a particle size of 0 to infinity for the grid box volume. In this model particle size is expressed in terms of its mass. An introduction to (bulk) microphysics schemes in general was added to the introduction.

Line 104. Remove "Actually".

**Response:** Removed 'actually'.

Line 125. This needs a reference as well.

**Response:** Added a reference.

Line 135. Remove "huge".

**Response:** Removed 'huge'.

Line 140. Please clarify what this means.

**Response:** The section was removed.

Line 154. Should contact ice nucleation be included?

**Response:** The physics of contact freezing of water droplets is in general not well understood. Including the process in bulk microphysic schemes requires an explicit aerosol model [see, e.g., Hande et al 17. (2017)]. Previous model studies indicate that contact freezing is less effective than other freezing modes (Hande et al 17., 2017).

Line 200. Since competition between homogeneous and heterogeneous nucleation is a significant feature of cloud ice formation, it is not clear how this approach would represent real clouds.

**Response:** The competition between heterogeneous and homogeneous nucleation is implicitly included in the model. Heterogeneous nucleation (e.g. deposition nucleation) occurs at lower supersaturations (wrt ice) than the homogeneous nucleation threshold. Hence, ice from heterogeneous nucleation can suppress homogeneous nucleation by depleting supersaturation due to depositional growth. If the homogeneous nucleation threshold is still reached, the effect of pre-existing ice (including ice from heterogeneous nucleation) is taken into account as a reduced, fictitious updraft velocity. See Eq. (14) in the manuscript and Kärcher et al. (2006).

Line 210. This expression needs to be weighted by the size distribution of the aqueous solution droplets. In fact, is nhom limited at all by the available droplets?

**Response:** Kärcher et al. (2002) assume a mono-disperse size distribution for the aqueous solution droplets. The implementation of their homogeneous nucleation parametrization in the ice modes scheme (and also the standard two-moment scheme) also use this simplification. Internally, the number of homogeneously frozen solution droplets within a physical time step is capped by the

number concentration of aqueous solution droplets. However, even under ideal conditions only a small fraction of available solution droplets will freeze (Spichtinger and Gierens, 2009).

Line 244. This is a strong assumption that makes this a very idealized model. Please clarify how it might affect the results.

**Response:** Most bulk microphysics used in operational and research models do not utilize an explicit aerosol model and instead rely on prescribed INP profiles. Introducing an explicit aerosol model requires the initialization and constrainment of aerosol sources and sinks in the model domain, which increases the complexity and uncertainty greatly. Tracking depleted INP as an additional tracer serve to constrain heterogeneous nucleation while not adding the complexity of an explicit aerosol model.

Line 269. This only works for INP immersed within cloud droplets. It is not clear whether it is applied that way.

**Response:** Since the ice modes (as well as the standard two-moment) scheme do not include an explicit aerosol model, we can not track if an INP is immersed in a liquid droplet or not. As such we consider all INPs activated by the immersion freezing scheme to be immersed in a droplet.

The number concentration of INPs is highly variable and uncertain. Thus we can consider the uncertainty if a INP is immersed in a droplet or not as part of the uncertainty of active INP in general. In a future study we will investigate the sensitivity of the ice mode schemes to the number concentration of profile of INPs.

Line 283. There is an issue here since "na" for deposition and immersion is different (i.e., whether they are inside/outside of cloud droplets).

**Response:** See response above.

Line 344. Why is this not dependent on the mass of other cloud species?

**Response:** The number concentration of secondary ice from rime splintering is depended on the rimed mass. Hence the amount of cloud droplet mass converted to graupel and hail. This includes riming of ice, snow, graupel and hail. We added a small section for clarification.

Liner 440. How many new tracers are added to the dynamics scheme? What is the domain of the simulation?

**Response:** The standard double-moment bulk scheme in ICON (Seifert and Beheng, 2006) uses 14 tracers. The new ice mode schemes uses an additional 8 tracers. Information to the domain and general model setup was expanded in the (new) model setup section (3.1).

Figure 3: Are these lines averaged over the whole domain?

**Response:** Yes they are the domain averages of the respective Ice Water Paths. We added a clarification in the text and figure labels.

Line 477, 497, 498 and other places. Using words like "weak", "stronger", "catches up" is ambiguous. Be precise on the description.

**Response:** We added quantitative statements to several parts in the results section for more precise descriptions.

Line 477. Not sure what the mass density means here as it has not been used to this point.

**Response:** Changed all instances of 'mass density' to 'mass content' for consistency.

Figure 5. Are these zonal means? On what domain?

**Response:** Figures 5. to 8. show vertical slices of the convective cloud through the center of the domain. Additional information was added in the text and figure captions for clarification.

Line 517. Remove "in an altitude".

**Response:** Removed 'in an altitude'.

Line 531. This separation is by design and may not occur in real clouds.

**Response:** In real clouds ice particles also originate from a distinct formation process. If we could track individual ice particles from their formation onward in real clouds (or even just in experiments), we would be able to apply the same classification as in this model.

Line 537. Correct "cloude".

**Response:** Corrected misspelling.

Line 539. Repeated "are".

**Response:** Corrected.

Line 544. Correct the units.

**Response:** Units were corrected.

Line 551. Please elaborate on this, not sure why there would be an increased number of collisions.

**Response:** With the addition of new cloud ice classes, the number of collision processes contributing to snow have increased. There are 4 additional calls to ice self collection and 10 new calls to collisions between ice modes. The equations for the ice-ice collision rates are in general non-linear. We added additional information concerning this topic in Section 2.5.2..

Line 557. What bias?

**Response:** The bias for the ice mode schemes to have higher aggregation rates than the base SB scheme at least in the environment of a deep convection cloud. However, the referred sentence was removed in the revised manuscript.

Line 581. Use "concentration"

**Response:** Changed all instances of 'number density' to 'number concentration' for consistency.

Line 589. Correct "then"

**Response:** Fixed misspelling.

Section 4: please add more discussion on the highly idealized nature of the setup and its implications (see general comment).

**Response:** We added some text for clarifying this issue in Section 4 (Discussion and summary).

Line 648-650. What is the ground truth here?

**Response:** The validation of the ice modes scheme in this study is done in Section 4.2 by comparison to the (unmodified) Seifert and Beheng (2006) double-moment scheme (SB06), which

is widely used in cloud research. The SB06 scheme uses the same assumptions for the heterogeneous  ice nucleation schemes and only differs in the number of cloud ice classes. Both the SB06 and ice modes scheme agree well with regards to evolution of all hydrometeor classes except snow.

**References**

Hande, L. B., Engler, C., Hoose, C., and Tegen, I.: Seasonal variability of Saharan desert dust and ice nucleating particles over Europe, Atmospheric Chemistry and Physics, 15, 4389–4397, https://doi.org/10.5194/acp-15-4389-2015, 2015.

Hande, L. B., C. Hoose, and C. Barthlott, 2017: Aerosol- and Droplet-Dependent Contact Freezing: Parameterization Development and Case Study. *J. Atmos. Sci.,* **74**, 2229–2245, https://doi.org/10.1175/JAS-D-16-0313.1.

Hulburt, H. M. and Katz, S.: Some problems in particle technology: A statistical mechanical formulation, Chemical engineering science, 19, 555–574, 1964

Kärcher, B. and Lohmann, U.: A parameterization of cirrus cloud formation: Homogeneous freezing of supercooled aerosols, Journal of Geophysical Research: Atmospheres, 107, AAC–4, https://doi.org/10.1029/2001JD000470, 2002

Kärcher, B., Hendricks, J., and Lohmann, U.: Physically based parameterization of cirrus cloud formation for use in global atmospheric models, Journal of Geophysical Research: Atmospheres, 111, https://doi.org/10.1029/2005JD006219, 2006.

Seifert, A. and Beheng, K. D.: A two-moment cloud microphysics parameterization for mixed-phase clouds. Part 1: Model description, Meteorology and atmospheric physics, 92, 45–66, https://doi.org/10.1007/s00703-005-0112-4, 2006.

Spichtinger, P. and K. M. Gierens (2009). "Modelling of cirrus clouds – Part 1a: Model description and validation". In: Atmospheric Chemistry and Physics.

Wernli, H., Boettcher, M., Joos, H., Miltenberger, A. K., and Spichtinger, P.: A trajectory-based classification of ERA-Interim ice clouds in the region of the North Atlantic storm track, Geophysical Research Letters, 43, 6657–6664, https://doi.org/10.1002/2016GL068922, 2016.